# Perceived sensorimotor synchrony enhances pain modulation and attenuates laser-evoked potentials
Xinyu Pan[1,2], Yian Xiao[1,2], Li Hu [1,2] & Xuejing Lu [1,2] ✉

Sensorimotor synchronization to music, referring to the temporal alignment of movement with auditory rhythms, has been associated with immersive engagement and enhanced pleasure. In the present study, we investigated its analgesic effects by three main experiments involving 224 healthy participants. Pain modulation was assessed by changes in responses to noxious laser stimuli before and after auditory stimulation (listening vs. no listening) and drumming activity (drumming vs. no drumming). Participants in the drumming-and-listening group exhibited greater reductions in pain intensity and unpleasantness than those in other groups, highlighting the analgesic advantage of such combination (Experiment 1). We then manipulated the perceived synchrony and revealed that participants in the in-phase synchrony group reported stronger perceived synchrony and greater pain reduction, when compared with asynchrony group (Experiment 2). Electrophysiological data further associated this analgesic effect with reduced laser-evoked N2 amplitudes (Experiment 3). Our findings indicate that perceived sensorimotor synchrony facilitates pain reduction, suggesting that rhythm-based interventions are a promising non-pharmacological approach to pain management.

Pain is far more than a simple physiological signal. It is a complex experience shaped by the interaction of sensory perception, cognition, emotion, and motor engagement[1,2]. This complicated relationship highlights the possibility to modulate pain experience through cognitive and emotional mechanisms[3,4], leading to a surge of interest in psychological approaches to pain management[5], such as cognitive behavioral therapy[6], mindfulness meditation[7], distraction strategies[8], and emotional regulation[9,10], offering non-invasive adjuncts and even alternatives to traditional pharmacological interventions. Among these psychological approaches, music has emerged as a particularly promising candidate for pain relief, as it offers psychophysiological benefits rapidly with minimal resource demands[11,12]. Far from a recent discovery, the therapeutic power of music, such as soothing discomfort and promoting healing, has been acknowledged for centuries, from the temples of ancient Egypt to the amphitheaters of Greece[13]. Numerous empirical studies over the past few decades have reinforced this long-standing belief, demonstrating that even passive music listening effectively alleviates various pain conditions, including acute, chronic, and post-operative pain[14–16].

Yet, music is not only heard but also felt. Humans exhibit an innate inclination to synchronize their movements with musical rhythms, evident in spontaneous behaviors such as swaying or head-nodding along to music[17,18]. This phenomenon (i.e., sensorimotor synchronization) reflects a coordination of rhythmic movement with an external auditory rhythm[19], which involves not only the objective temporal alignment between movement and auditory beat, but also the subjective perception of such alignment (i.e., perceived synchrony). Indeed, when movements such as tapping or drumming are perceived to align with musical beats, the brain treats it as a match between predicted and actual sensory inputs. Consistent with the framework of predictive coding, this process is rewarding and may lead to emotional engagement and pleasurable experience[20,21]. Therefore, it has been suggested that perceived synchrony is more closely associated with affective and motivational responses than objective temporal alignment alone. For example, individuals' perceived motor synchrony with a musical beat is more likely to predict the groove experience (i.e., the pleasurable urge to move) than objective measured synchrony[22]. This affective engagement may further contribute to the analgesic effects through the recruitment of reward and opioid systems[23–25]. Supporting this, a recent study demonstrated that music tuned to match participants' spontaneous production rates, an internal rhythm reflecting individual motor preferences, produced stronger reductions in pain compared to faster or slower tempos[26]. This is possibly because such tempo alignment likely enhances the subjective feeling of synchrony, thereby reinforcing the pleasurable and immersive qualities of rhythmic engagement.

[1]State Key Laboratory of Cognitive Science and Mental Health, Institute of Psychology, Chinese Academy of Sciences, Beijing, China. [2]Department of Psychology, University of Chinese Academy of Sciences, Beijing, China. ✉e-mail: luxj@psych.ac.cn

In addition to the emotional impact, sensorimotor synchronization is thought to stabilize one's internal representation of rhythms[27], as it involves a dynamic calibration mechanism with temporal adaptation and predictive anticipation, which further facilitates sensory integration and cognitive engagement[28]. It has been suggested that, under specific circumstances, sensorimotor synchronization to musical rhythm may induce an immersive state characterized by focused attention and reduced self-consciousness[29]. In this state, individuals usually experience heightened concentration and reduced fatigue[30], both of which are associated with pain modulation[31,32]. From this perspective, sensorimotor synchronization may exhibit stronger analgesic effects than music or movement alone, engaging distinct mechanisms beyond those underlying music-induced and movement-induced hypoalgesia[25].

Despite promising findings, the precise mechanisms through which sensorimotor synchronization modulates pain perception remain unclear. Although behavioral evidence has shown that rhythmic movement to music can reduce pain[24], it is unclear which specific components of synchronization drive this effect. In this study, we focused specifically on the subjective aspect of sensorimotor synchronization. By manipulating the temporal alignment between auditory and visual cues, where the visual cues guided participants' movement, we aimed to induce various subjective experiences of synchrony with rhythmic auditory input, rather than to measure synchronization accuracy directly. Based on the framework of predictive coding and prior findings, we hypothesized that pain reduction would be greatest when movements were perceived as temporally aligned with auditory input. Specifically, we expected that conditions facilitating perceived in-phase synchrony would yield stronger analgesic effects than either listening or drumming alone, or non-synchronous pairings. In addition, accumulating evidence from electroencephalographic (EEG) studies indicates that both music listening and rhythmic movement can modulate brain responses to nociceptive stimuli[2,33]. For example, consistent reductions in the amplitudes of laser-evoked components, such as the N2 and P2, have been observed following music listening or motor engagement, reflecting attenuated central nociceptive processing[2,33]. Therefore, we predicted that in-phase synchrony would be associated with reduced amplitudes of related components.

Here, by combining music and movement, the present study aims to investigate whether coordinated rhythmic activities can reduce pain and, if so, to elucidate the cognitive and emotional mechanisms through which such effects occur. Across three experiments with 224 healthy participants, we found that combining active drumming with rhythmic auditory sequences led to the most significant reductions in perceived pain intensity and unpleasantness. Crucially, this analgesic effect was modulated by the subjective experience of temporal alignment, with in-phase synchrony producing greater pain reduction than asynchrony. In addition, perceived in-phase sensorimotor synchrony was associated with attenuated laser-evoked N2 amplitudes, and mediation analysis indicated that these changes in N2 partially mediated the observed pain reduction. Furthermore, an exploratory supplementary experiment revealed that perceived sensorimotor synchrony may modulate pain processing by altering spontaneous brain oscillations (see Supplementary Experiment). These findings highlight that perceived sensorimotor synchrony plays an important role in modulating the neural processing of pain, thereby suggesting rhythm-based interventions as a promising non-pharmacological approach for pain management.

## Results

### Experiment 1: combined effects of listening and drumming on pain perception

We first investigated the individual and combined effects of rhythmic auditory sequences (percussion excerpts) and active drumming (on an electronic drum pad guided by visual cues) on pain perception across four groups: silence, drumming-only, listening-only, and drumming-and-listening (see Fig. 1A). Pain was induced by noxious laser stimuli. The effects of these interventions were assessed by comparing pre-task and post-task changes in pain intensity, unpleasantness, as well as positive and negative affect (see Fig. 1B).

As shown in Fig. 2A, a significant interaction was observed for changes in pain intensity ($F_{(1,76)} = 10.64$, $P = 0.002$, $\eta_p^2 = 0.12$). Further analyses revealed that reductions in pain intensity were comparable for the drumming and no-drumming groups in the absence of listening (drumming-only vs silence: $P = 0.66$). However, a significantly greater reduction in pain intensity was observed when participants were drumming to percussion excerpts, compared to those who were only listening (drumming-and-listening vs. listening-only: $P < 0.001$), suggesting that the effect of drumming on pain relief was dependent on the presence of percussion excerpts. In addition, participants in the two listening groups experienced greater reductions in pain intensity than those in the two no-listening groups, regardless of whether they engaged in drumming (drumming-and-listening vs. drumming-only: $P < 0.001$) or not (listening-only vs. silence: $P = 0.023$). Furthermore, significant main effects of listening ($F_{(1,76)} = 42.77$, $P < 0.001$, $\eta_p^2 = 0.36$) and drumming ($F_{(1,76)} = 15.11$, $P < 0.001$, $\eta_p^2 = 0.12$) were found for changes in pain intensity.

A significant main effect of listening was observed for changes in unpleasantness ($F_{(1,76)} = 29.51$, $P < 0.001$, $\eta_p^2 = 0.28$), suggesting that the two listening groups experienced greater reductions in unpleasantness than the two no-listening groups. However, there was no significant main effect of drumming ($F_{(1,76)} = 3.67$, $P = 0.059$), nor a significant interaction effect between listening and drumming for changes in unpleasantness ($F_{(1,76)} = 3.47$, $P = 0.066$).

In addition, significant effects of listening were observed for changes in positive ($F_{(1,76)} = 8.03$, $P = 0.006$, $\eta_p^2 = 0.10$) and negative ($F_{(1,76)} = 15.90$, $P < 0.001$, $\eta_p^2 = 0.17$) affect (Fig. 2B), indicating that the two listening groups experienced significant increases in positive affect and decreases in negative affect, compared to the two no-listening groups. Moreover, neither the main effect of drumming ($F_{(1,76)} = 2.66$, $P = 0.11$) nor its interaction with listening ($F_{(1,76)} = 2.82$, $P = 0.10$) was statistically significant for positive affect. Similar null results were found for negative effect ($F_{(1,76)} = 0.23$, $P = 0.63$ and $F_{(1,76)} < 0.001$, $P = 1.00$, respectively).

### Experiment 2: perceived sensorimotor synchrony modulates pain

Building upon the analgesic benefits observed in Experiment 1, Experiment 2 further investigated how varying degrees of perceived sensorimotor synchrony modulated pain perception. Participants drummed following visual cues, while the temporal alignment between auditory drumbeats and these visual cues was manipulated to create three distinct stimulus conditions: in-phase synchrony (0° phase shift), anti-phase synchrony (180° phase shift), and asynchrony (mismatched tempos; see Fig. 1C). To check the effectiveness of this manipulation, we collected the perceived synchrony (Q1), auditory-cue reliance (Q2), visual-cue reliance (Q3), and task performance satisfaction (Q4) scores for each stimulus condition.

As shown in Supplementary Fig. 1, significant group differences were found for Q1 (perceived synchrony) and Q2 (auditory-cue reliance) scores ($H_{(2)} = 34.95$, $P < 0.001$; $H_{(2)} = 20.60$, $P < 0.001$; respectively), suggesting that participants' perceived synchrony and reliance on auditory cues varied systematically with the assigned stimulus condition. Specifically, Q1 scores were highest in the in-phase synchrony group compared to both the anti-phase and asynchronous groups ($P < 0.001$ and $P = 0.004$, respectively); the asynchrony group also scored higher than the anti-phase synchrony group ($P = 0.020$). For Q2, participants in the in-phase synchrony and asynchrony groups relied more on auditory cues than those in the anti-phase group ($P < 0.001$ and $P = 0.038$, respectively). In contrast, no significant group differences were observed for Q3 (visual-cue reliance) scores ($H_{(2)} = 3.11$, $P = 0.21$), suggesting all groups similarly attempted to follow the visual flashing-dot cues as instructed. Furthermore, Q4 scores (task performance satisfaction) also showed no significant differences across groups ($H_{(2)} = 4.26$, $P = 0.12$), indicating comparable subjective task performance across conditions and not influenced by the type of sensorimotor synchronization task. Taken together, these results confirm that the

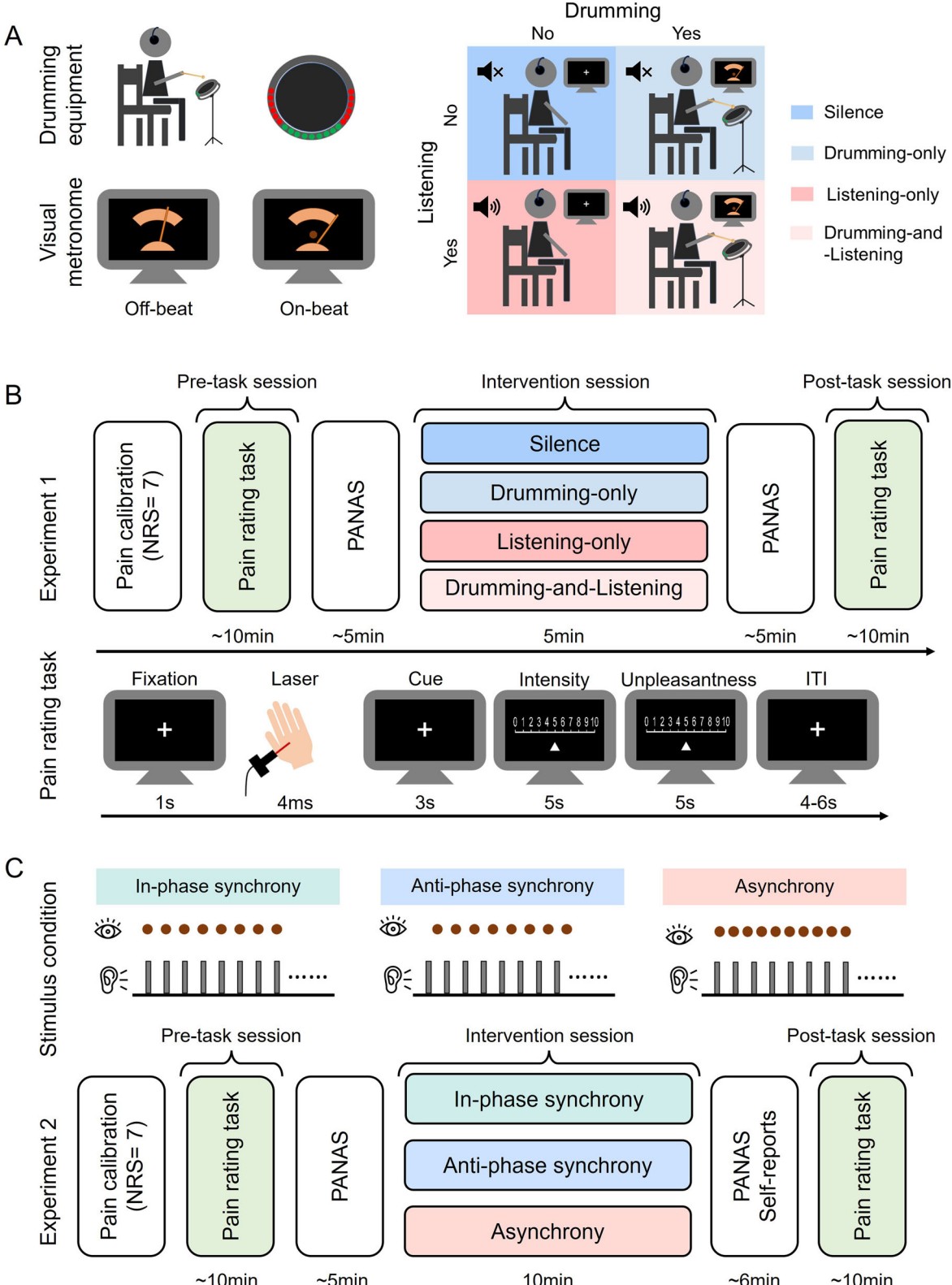

**Fig. 1 | Stimuli and experimental design of Experiments 1 and 2. A** An illustration of drumming equipment and the visual metronome in the video (left), as well as four experimental groups in Experiment 1 (right). **B** The procedure of Experiment 1 with a representative of a single trial in the pain rating task. **C** The procedure of Experiment 2 with an illustration of the three stimulus conditions (in-phase synchrony, anti-phase synchrony, and asynchrony). NRS numerical rating scale, PANAS Positive and Negative Affect Scale, ITI intertrial interval.

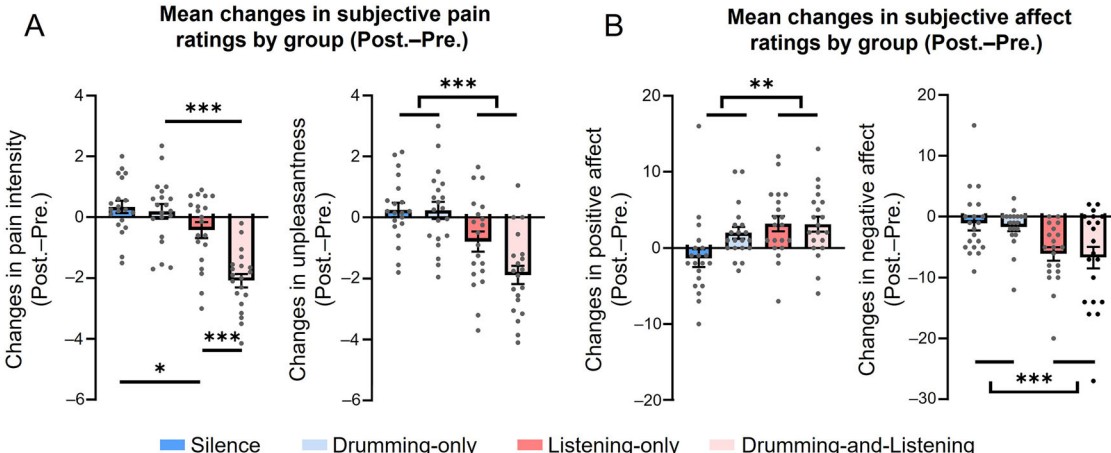

**Fig. 2 | Results of Experiment 1 ($n = 20$ for each group; total $N = 80$).**
**A** Comparisons of changes in pain intensity and unpleasantness following the presence (red) or absence (blue) of percussion excerpts, during conditions where participants were instructed to drum (lighter colors) or not to drum (darker colors). For changes in pain intensity, the listening-and-drumming group showed a more pronounced reduction; for changes in unpleasantness, the two listening groups showed greater reductions than the two drumming groups. **B** Comparisons of changes in positive and negative affect. Positive affect significantly increased and negative affect significantly decreased in the groups involving listening than in those not. Error bar represents ±one standard error of the mean; Pre. the pre-task session, Post. the post-task session. ***$P < 0.001$; **$P < 0.01$; *$P < 0.05$.

manipulation successfully induced distinct experiences of synchrony and cue reliance across conditions, while visual cue tracking and task satisfaction remained stable.

Regarding pain perception, significant group effects were observed for changes in pain intensity ($F_{(2,63)} = 31.68$, $P < 0.001$, $\eta_p^2 = 0.50$) and unpleasantness ($F_{(2,63)} = 6.82$, $P = 0.002$, $\eta_p^2 = 0.18$), as shown in Fig. 3A. Post hoc pairwise comparisons revealed that reductions in pain intensity were significantly greater in both the in-phase synchrony and anti-phase synchrony groups than in the asynchrony group (both $P < 0.001$). However, reductions in unpleasantness were significantly greater only in the in-phase synchrony group and not the anti-phase synchrony group, when compared to the asynchrony group ($P = 0.002$ and $P = 0.089$, respectively). Furthermore, no group differences were found for changes in positive affect ($F_{(2,63)} = 1.81$, $P = 0.17$) or negative affect ($F_{(2,63)} = 0.14$, $P = 0.87$).

We further examined the link between subjective pain ratings and self-report measures collected during the manipulation check. Correlation analyses revealed that changes in pain ratings were negatively correlated with the degree to which participants perceived synchrony (Q1; pain intensity: $\rho = -0.53$, $P < 0.001$; unpleasantness: $\rho = -0.27$, $P = 0.028$), as well as with the extent to which they relied on auditory drumbeats for drumming (Q2; pain intensity: $\rho = -0.39$, $P = 0.001$; unpleasantness: $\rho = -0.25$, $P = 0.040$; Fig. 3B). In other words, the more participants felt synchronized and relied on rhythmic auditory sequence, the greater their reductions in pain ratings. In contrast, changes in pain ratings were not significantly correlated with the extent to which they relied on flashing dots for drumming (Q3; pain intensity: $\rho = 0.03$, $P = 0.83$; unpleasantness: $\rho = -0.10$, $P = 0.44$) and their satisfaction of task performance (Q4; pain intensity: $\rho = -0.14$, $P = 0.26$; unpleasantness: $\rho = -0.15$, $P = 0.23$).

### Experiment 3: neural correlates of perceived synchrony in pain modulation

Based on the behavioral evidence from Experiments 1 and 2, Experiment 3 further elucidated the neural mechanisms underlying pain modulation by perceived sensorimotor synchrony using EEG, with laser-evoked potentials (LEPs) recorded from scalp electrodes (Fig. 4). Pain was induced and assessed as in previous experiments, while LEPs were analyzed for N1, N2, and P2 components.

Consistent with our behavioral findings, we observed a greater decrease in laser-evoked N2 amplitude in the in-phase synchrony group compared to the asynchrony group. As shown in Fig. 5A, significant main effects of group were found on changes in pain intensity ($F_{(2,75)} = 4.05$, $P = 0.021$, $\eta_p^2 = 0.10$) and unpleasantness ($F_{(2,75)} = 3.64$, $P = 0.031$, $\eta_p^2 = 0.09$). Post hoc pairwise comparisons revealed that reductions in pain intensity and unpleasantness were significantly greater in the in-phase synchrony group when compared with the asynchrony group (pain intensity: $P = 0.018$; unpleasantness: $P = 0.043$). This effect aligns with the LEP findings, where there was a significant group difference for changes in N2 amplitude but not in P2 amplitude ($F_{(2,75)} = 3.94$, $P = 0.024$, $\eta_p^2 = 0.10$; and $F_{(2,75)} = 0.07$, $P = 0.94$, respectively; Fig. 5A, B). Specifically, while the in-phase group showed significant reductions compared to asynchrony ($P = 0.048$), no significant differences were found between the in-phase and anti-phase groups ($P = 0.058$), nor between the anti-phase and asynchrony groups for changes in N2 amplitude ($P = 1.00$). Furthermore, changes in N1 amplitude, and in the latencies of N1, N2, and P2, did not reveal any significant group differences (all $P > 0.05$; see Supplementary Fig. 2).

Further analysis on the link between subjective pain ratings and laser evoked brain responses indicated that changes in N2 amplitude correlated negatively with changes in pain intensity ($\rho = -0.38$, $P < 0.001$; Fig. 5C), but not significantly with unpleasantness ($\rho = -0.18$, $P = 0.12$), whereas correlations between changes in P2 amplitude and pain ratings were not statistically significant (pain intensity: $\rho = 0.20$, $P = 0.08$; unpleasantness: $\rho = 0.20$, $P = 0.074$). The mediation analysis confirmed that changes in N2 amplitude significantly mediated the effect of group on subjective pain ratings (Fig. 5D). Using the asynchrony group as a reference, changes in N2 amplitude significantly mediated the effect on changes in pain intensity ($a_1*b = -0.20$, SE $= 0.10$, 95% confidence interval (CI) $= [-0.40, -0.02]$, $\beta = -0.11$, $P = 0.049$), with a significant direct effect remaining after including the mediator in the in-phase synchrony group ($c_1' = -0.44$, SE $= 0.20$, 95% CI $= [-0.81, -0.07]$, $\beta = -0.25$, $P = 0.020$). In contrast, there was no significant mediating effect ($a_2*b = -0.01$, SE $= 0.08$, 95% CI $= [-0.20, 0.13]$, $\beta = -0.003$, $P = 0.94$) or direct effect ($c_2' = -0.26$, SE $= 0.24$, 95% CI $= [-0.72, 0.22]$, $\beta = -0.15$, $P = 0.28$) in the anti-phase synchrony group. The results suggest that reductions in pain intensity in the in-phase synchrony group were mediated by changes in N2 amplitude. Further neurophysiological findings regarding the modulation of spontaneous brain oscillations are presented in the Supplementary Experiment.

### Discussion

In this study, we examined the effects of perceived sensorimotor synchrony on pain perception. Our results demonstrated that the combination of

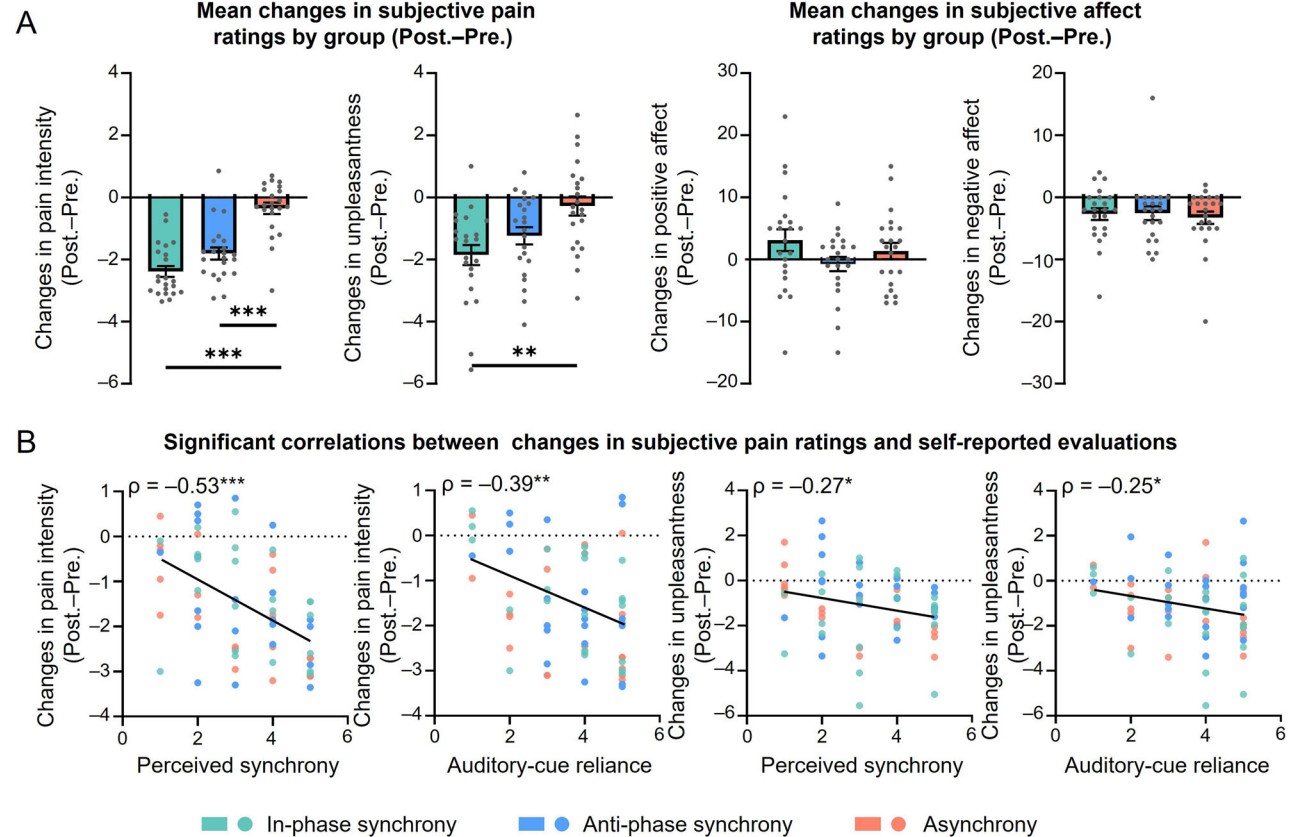

**Fig. 3 | Results of Experiment 2 ($n = 22$ for each group; total $N = 66$).**
**A** Comparisons of changes in pain ratings and affect among in-phase synchrony (green), anti-phase synchrony (blue), and asynchrony (red) groups. Participants in the in-phase synchrony group showed greater reduction in pain intensity and unpleasantness compared to the asynchrony group. **B** Significant correlations between changes in pain ratings (both pain intensity and unpleasantness) and the degree to which participants felt the synchrony as well as the extent to which they relied on auditory drumbeats for drumming. Error bar represents ±one standard error of the mean; Pre. the pre-task session, Post. the post-task session. ***$P < 0.001$; **$P < 0.01$; *$P < 0.05$.

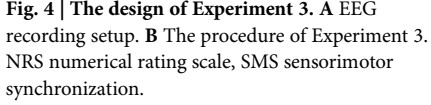
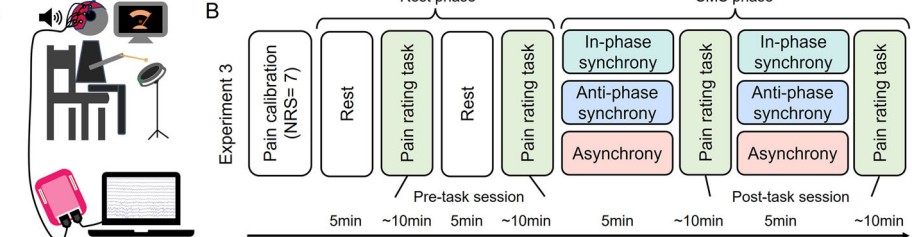

**Fig. 4 | The design of Experiment 3. A** EEG recording setup. **B** The procedure of Experiment 3. NRS numerical rating scale, SMS sensorimotor synchronization.

rhythmic auditory sequences with active drumming led to the greatest reductions in pain intensity and unpleasantness (Experiment 1). Importantly, participants' subjective sense of temporal alignment between their drumming and the auditory rhythm (i.e., perceived synchrony) was positively correlated with the magnitude of pain reduction (Experiment 2). At the neurophysiological level, the in-phase synchronization led to a significant attenuation of the LEP N2 component. Mediation analysis further reveals that reduced N2 amplitude partially associated with the relationship between perceived synchrony and pain reduction (Experiment 3). Furthermore, in an exploratory investigation detailed in the Supplementary Experiment, we observed that in-phase sensorimotor synchronization was associated with modulations of spontaneous brain oscillations, including modulations in α-band and high γ-band activity, and changes in the aperiodic exponent. These findings indicated that the perceived temporal alignment between action and sound plays an important role in modulating both the perception and neural processing of pain. By demonstrating the

importance of perceived synchrony in pain modulation, our study suggests the potential of sensorimotor synchronization in pain management.

One of the main findings of this study is that combining rhythmic auditory sequences with active drumming led to a more pronounced reduction in perceived pain intensity in response to noxious stimuli, compared with other groups (Experiment 1). This aligns with recent research showing that participants experienced less pain when actively tapping their feet to music but not simply listening to music, compared to tapping in silence[24]. However, it cannot exclude the impact of attentional capture, as pain stimulation was applied during the experimental manipulation. By contrast, our study adopted a pre–post design to minimize the influence of differences in attentional demands across groups in the observed effects[12,34]. Supported by further evidence that active musical engagement (e.g., singing, dancing, and drumming) elevated pain threshold and increased positive affect[35], these findings highlight the benefits of active, embodied interaction with rhythm in music-based pain intervention.

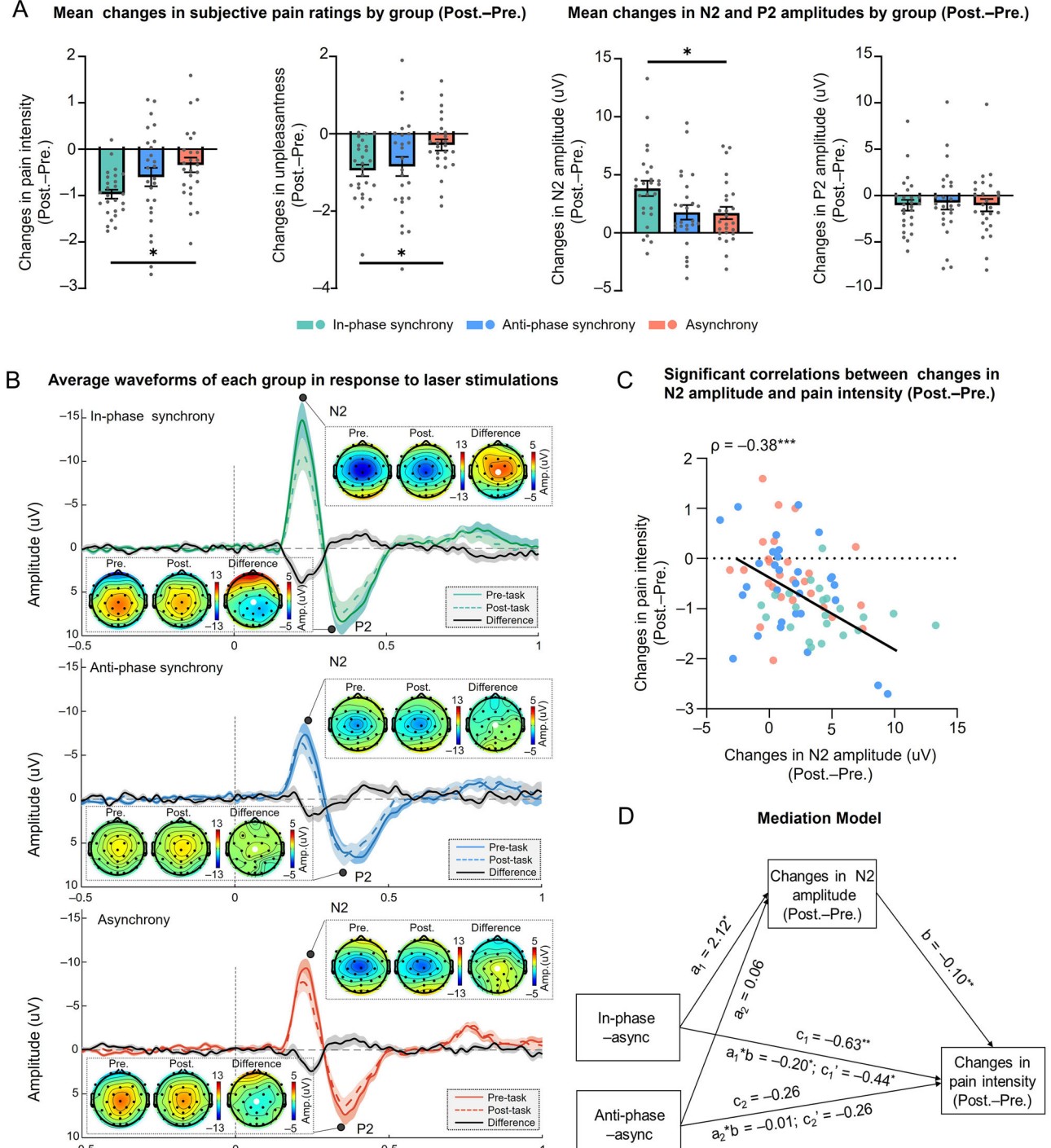

**Fig. 5 | Results of Experiment 3 ($n = 26$ for each group; total $N = 78$).**
**A** Comparisons of changes in subjective pain ratings as well as N2 and P2 amplitudes among the in-phase synchrony (green), anti-phase synchrony (blue), and asynchrony (red) groups. **B** Average waveforms at Cz electrode (with bilateral mastoid as the reference electrode) of each group in response to laser stimulations in the pre-task (solid line) and the post-task (dash line) sessions (black lines were difference waves between two sessions), as well as the group-level scalp topographies of N2 (upper) and P2 (lower) components at their respective peak latencies. **C** Significant correlation between changes in N2 amplitude and changes in pain intensity. **D** Changes in N2 amplitude mediated the effect of group on pain intensity ratings. The coefficients for each path are presented. Error bar represents ±one standard error of the mean. Pre. the pre-task session, Post. the post-task session. ***$P < 0.001$; **$P < 0.01$; *$P < 0.05$.

When individuals are required to drum along with music, they need to track the beat, move in synchrony, and process proprioceptive and auditory feedback from their body and musical instruments[36]. In other words, they have to maintain rhythmic accuracy and timing, which involves precise motor control and the coordination of auditory and motor systems[37]. This process is often accompanied by enhanced cognitive engagement and attentional focus[30], promoting flow-like states characterized by deep immersion and effortless absorption[29]. Although the present study did not directly measure the state of flow, the drumming-and-listening activity is consistent with conditions previously linked to flow states, such as repeated

rhythmic patterns, embodied musical interaction, and motor predictability[38]. The strong connection between music and the body is well established, spanning diverse musical styles and contexts[39]. Evidence from neuroimaging studies further suggests that the urge to move to music (i.e., groove) is associated with increased activity in the reward system, such as the nucleus accumbens[40]. Therefore, drumming to music, a form of rhythmic activity that has been shown to evoke the sensation of groove under certain circumstances, is expected to contribute to positive emotional states and pain relief[17]. These mechanisms may jointly contribute to the observed analgesic benefits by reinforcing pleasurable engagement and modulating affective-motivational components of pain processing.

Our manipulation check performed in Experiment 2 indicated a successful manipulation of perceived synchrony, in which participants in the in-phase synchrony condition reported significantly stronger perceived audio-visual synchrony. The analgesic effect induced by drumming to music is strongly modulated by the degree of synchrony between auditory drumbeats and percussion strokes (i.e., phase differences of 0°, 180°, and random), as evidenced by the progressive decrease in both pain intensity and unpleasantness with increasing levels of subjective ratings of perceived synchrony. Furthermore, participants in three groups did not differ in terms of their scores of visual cues reliance and performance satisfaction, suggesting that the observed pain relief was not simply due to differential task ease or general satisfaction across groups.

Perceived synchrony may alleviate pain by reducing multisensory conflict. When auditory cues and motor actions are perceived as synchronized, the brain integrates sensory information more efficiently, which decreases cognitive dissonance and reduces the need for conflict resolution between incongruent sensory and action signals[41]. In contrast, the asynchrony condition, where there is a mismatch between auditory beats and motor actions, generates sensory dissonance and increases cognitive load. This interference is likely to disrupt the brain's ability to achieve optimal sensory integration[42]. These results emphasize the importance of temporal alignment with auditory rhythms and motor actions to optimize pain modulation[30]. In line with this idea, we observed significant correlations between the subjective sense of synchrony and reductions in both pain intensity and unpleasantness. These findings suggest that the greater perceived synchrony was associated with stronger analgesic effects.

Consistent with behavioral findings, we observed a larger decrease in laser-evoked N2 amplitude in the in-phase synchrony group than in the asynchrony group. The N2 component, evoked by noxious stimuli, is primarily associated with the activation of the bilateral operculo-insular cortex and secondary somatosensory cortex. The N2 component is highly associated with pain perception[43,44], and believed to be involved in the initial sensory processing of painful stimuli, playing a significant role in the cognitive evaluation and preparatory responses to pain[45]. More importantly, correlation and mediation analyses suggest that changes in N2 component were significantly associated with reductions in pain ratings, suggesting that N2 may serve as a neurophysiological correlate of pain modulation.

Furthermore, perceived sensorimotor synchrony may modulate pain processing by altering spontaneous brain oscillations. Exploratory results from a supplementary experiment revealed that in-phase synchrony was tentatively associated with increased high γ-band oscillations, a decreased aperiodic exponent, and sustained α-band activity over posterior regions when compared to a rest condition. Although the experimental design does not fully isolate synchrony-specific effects from those of general rhythmic movement, these observations align with studies linking such oscillatory changes to enhanced cognitive engagement and a shift in cortical excitability[46–48], which have in turn been associated with pain reduction[49]. A more comprehensive discussion of its findings and limitations is provided in the Supplementary Experiment.

Although the present study provides insights into the role of perceived sensorimotor synchrony in pain modulation, several research questions related to this study need to be addressed in further investigations. First, although our manipulation check confirmed that participants subjectively perceived and responded to different levels of synchrony and maintained comparable task satisfaction, we did not collect objective measures of synchronization accuracy (e.g., precise temporal alignment of drumming strokes to the beat). Therefore, while we demonstrated the impact of perceived synchronization, the specific contribution of objective sensorimotor coupling accuracy to the observed analgesic effect, independent of perceived ease or fluency, warrants further investigation with objective measures of performance. Second, an exploratory investigation into spontaneous brain oscillations was conducted in addition to the behavioral and LEP measures reported in three main experiments and is detailed in Supplementary Experiment. This supplementary experiment provides preliminary insights. However, due to its limitations, such as the absence of a non-synchronous motor activity control group, we cannot reach definitive conclusions about the unique contribution of perceived synchrony to these oscillatory changes independent of general motor activity. Third, since no validated psychological instruments were employed to directly assess subjective flow experience, our inferences regarding the induction of an immersive state in the supplementary experiment should be interpreted cautiously. Fourth, future studies should further examine the clinical relevance of these results by investigating whether other forms of rhythmic synchronization, such as dancing and coordinated group activities, yield similar analgesic effects. It would also be important to assess the generalizability of these effects across diverse populations, including those with chronic pain conditions, as well as to identify optimal stimulation parameters (e.g., tempo, rhythm complexity) for maximizing pain relief. Fifth, longitudinal studies are necessary to evaluate the long-term effects of rhythmic synchronization training on pain modulation.

In summary, this study demonstrates that perceived sensorimotor synchronization, particularly the in-phase synchrony, plays a significant role in the modulation of pain perception. These findings advance our understanding of the neurophysiological mechanisms by which the perceived synchrony affects pain processing. By clarifying how coordinated rhythmic activities reduce pain, our work highlights the potential of rhythm-based interventions for pain management.

## Methods
### Participants
We conducted three main experiments with a total of 224 healthy participants to investigate the analgesic effects of perceived sensorimotor synchrony. An additional exploratory experiment examining spontaneous brain oscillations is detailed in Supplementary Experiment. To eliminate potential carryover effects, each participant took part in only one experiment. All participants were screened by self-reports to ensure the absence of psychiatric disorder, sensory abnormalities, or neurological conditions. None reported chronic pain or the use of analgesics within 24 h prior to participation. The study protocols were approved by the Ethics Committee of the Institute of Psychology, Chinese Academy of Sciences (ID: H21103), and all participants signed written informed consent upon arrival. All ethical regulations relevant to human research participants were followed. To minimize expectancy effects, participants were kept unaware of the study's hypothesis regarding pain modulation. Additionally, the experimenter who administered the pain rating tasks was blinded to the experimental conditions.

Sample sizes for each experiment were determined through a priori power analyses using G*Power software (available at http://www.Gpower.hhu.de/en.html)[50] with statistical power $(1 - \beta)$ set at 0.8, a large effect size $(f = 0.4)$ based on Werner et al.[24], and a significance level $(\alpha)$ at 0.05 (see individual experiments for details). Eighty healthy participants were recruited (46 females and 34 males, mean ± SD age = 23.8 ± 2.3 years, age range: 18–29 years) in Experiment 1. The power analysis for a 2 × 2 factorial analysis of variance (ANOVA), targeting the main effects and interactions, indicates a minimum required sample size of 52. Sixty-six participants were recruited (41 females and 25 males, mean ± SD age = 23.8 ± 2.4 years, age range: 19–30 years) in Experiment 2. The power analysis for ANOVA focusing on fixed effects across three groups indicated a minimum required sample size of 66. Seventy-eight participants were recruited (39 females and

39 males, mean ± SD age = 22.8 ± 2.4 years, age range: 18–27 years) in Experiment 3. The sample size was determined using the same a priori power analysis approach as in Experiment 2, ensuring adequate statistical power to detect significant effects.

## Stimuli and drumming equipment

In the main experiments, auditory stimuli consisted of two 2.5-min percussion excerpts extracted from a drum set tutorial. Both excerpts featured a common drumbeat (i.e., 4/4 simple straight beat) performed by the same player, with tempos set at 90 and 120 beats per minute (bpm), respectively. These tempos were selected based on two primary considerations: first, they fall within the empirically established optimal range for human sensorimotor synchronization, which is supposed to facilitate comfortable and natural rhythmic movement[19,51] and pain modulation[26]; second, the inclusion of two distinct tempos introduces temporal variability, which helps reduce monotony and maintain participants' engagement throughout the experiment. During the experiment, the 90-bpm excerpt was always presented first, followed by the 120-bpm excerpt. Corresponding muted video clips displayed a dot flashing in synchrony with a pendulum metronome, set to match the tempo of each excerpt (90 or 120 bpm). Each video lasted 2.5 min, following the same sequence as the auditory stimuli.

Nociceptive-specific radiant-heat stimuli were generated by an infrared neodymium yttrium aluminum perovskite laser (wavelength: 1.34 μm, pulse duration: 4 ms, Electronic Engineering, Italy). Laser pulses were directed to a squared area ($4 \times 4$ cm$^2$) on the dorsum of each hand, with the laser beam diameter at the stimulated site being 7 mm. After each stimulus, the laser beam target was shifted by at least 1 cm in a random direction within the predefined squared area to avoid nociceptor fatigue or sensitization. The energy of laser stimuli was individually calibrated by increasing the stimulus energy in steps of 0.25 J from 2 J in a pain calibration session. Participants were instructed to rate the intensity of pain elicited by each laser stimulus on their dominant hands until an average rating of 7 out of 10 was obtained on a numerical rating scale (NRS) ranging from 0 (no sensation) to 10 (the worst pain imaginable), which represents a moderate-to-severe level of pain. The mean laser energies were 3.63 ± 0.58 J with a range of 2.75–4.5 J in Experiment 1, 3.48 ± 0.41 J with a range of 2.75–4.5 J in Experiment 2, and 3.93 ± 0.41 J with a range of 3–4.5 J in Experiment 3.

The percussion instrument was an electronic drum pad (diameter: 25 cm; ENOY, China), mounted on a height-adjustable drum stand for individualized comfort. This setup allowed participants to easily strike the drum with a drumstick using their dominant hand while seated (Fig. 4A). The drum pad featured 29 built-in cue lights evenly distributed around its outer ring, providing real-time feedback on drumming force—more lights illuminated as the applied force increased.

## Experimental design and procedures

We employed a between-subjects design in Experiment 1, in which participants were randomly assigned to one of four experimental groups, defined by the presence or absence of auditory stimuli (listening vs. no listening) and drumming activity (drumming vs. no drumming). Participants in the silence group (i.e., neither listening nor drumming) were asked to focus on a fixation cross displayed at the center of a computer monitor and stay still. Participants in the listening-only group (i.e., listening but no drumming) were instructed to solely listen to the percussion excerpts without performing any movements, while those in the drumming-only group (i.e., drumming but no listening) were required to strike the electronic drum pad at the tempo demonstrated in the muted video without listening to any auditory stimulus. Those in the drumming-and-listening group (i.e., drumming and listening) were presented with both the percussion excerpts and videos simultaneously with aligned tempo and instructed to drum at the demonstrated tempo shown in the video. To ensure proficiency, participants in the drumming-only and drumming-and-listening groups practiced before the formal experiment. The practice session was terminated once participants could maintain the required tempo with moderate and stable

drumming force (approximately half of the drum pad's lights were illuminated).

The procedure of Experiment 1 is shown in Fig. 1B. Following a pain calibration session, participants underwent three main sessions: the pre-task, the intervention, and the post-task sessions. This experimental design aimed to minimize potential confounds related to cognitive load or emotional state during the intervention itself. To this end, pain rating tasks were conducted before and after the intervention, allowing us to evaluate the aftereffects of different interventions on pain perception without contamination from transient factors during the task. During the intervention session, participants in different groups received varying experimental manipulations, while in the pre-task and post-task sessions, participants across all groups completed an identical pain rating task. The task consisted of 20 trials, where each trial started with a 1-s fixation cross, followed by a laser stimulus of predefined intensity applied to the dorsum of either the right or left hand (10 stimuli per hand, pseudorandomized with no more than three consecutive stimuli on the same hand). After each laser stimulus, participants were prompted by visual cues lasting 3 s to rate pain intensity (no sensation–the worst pain imaginable) and unpleasantness (no unpleasantness–most unpleasantness) on the same 0–10 NRS within 10 s (5 s per rating). The intertrial interval (ITI) was 4–6 s randomly. To assess the emotional impact of the interventions, participants' emotional states were evaluated before and after the intervention session using the Chinese version of the Positive and Negative Affect Scale (PANAS)[52]. Throughout the experiment, participants wore headphones (Sennheiser HD201, Sennheiser Electronic, Germany), and when auditory stimuli were played, the sound intensity was set to a comfortable listening level.

We employed a between-subjects design in Experiment 2, in which participants were randomly assigned to one of the three experimental groups, each corresponding to a specific stimulus condition (in-phase synchrony, anti-phase synchrony, or asynchrony). The temporal alignment between auditory drumbeats and visual flashing dots (which guided participants' drumming movements) was manipulated to create these conditions. For instance, in the in-phase synchrony condition, drumbeats were temporally aligned with flashing dots; in the anti-phase synchrony condition, a stable time lag of one-eighth note (1/8) in a 4/4 rhythm resulted in a 180° phase shift; and in the asynchrony condition, a 90-bpm auditory excerpt was paired with a 120-bpm video clip (and vice versa) to produce unpredictable visual cues relative to drumbeats. This indirect modulation of motor synchronization with auditory rhythm, through audio-visual alignment, aimed to induce varying subjective experiences of synchrony. The procedure of Experiment 2 is shown in Fig. 1C. The formal experiment consisted of a pain calibration session followed by three main sessions. The same pain rating task as in Experiment 1 was administered during both pre-task and post-task sessions. To assess participants' perception of the stimuli and their engagement strategies, we administered a brief manipulation check questionnaire immediately after the intervention session in addition to the PANAS. Participants were asked to evaluate their perception of audio-visual synchrony (Q1: the extent to which the drumbeats and flashing dots were synchronized with each other), drumming strategies (Q2: the extent to which you were playing the drum in response to drumbeats; Q3: the extent to which you were playing the drum in response to flashing dots), and subjective task performance (Q4: the extent to which you were satisfied with your task performance) using a 5-point Likert scale.

We adopted a between-subjects design and used the same stimulus conditions in Experiment 3. Since we had already conducted a manipulation check in Experiment 2 confirming that participants perceived different levels of synchrony and used auditory and visual cues differently across conditions, we did not repeat the manipulation check in Experiment 3 to avoid redundancy. In addition, the procedure of Experiment 3 was modified to optimize EEG data quality. Specifically, EEG signals were recorded (Fig. 4A), and the number of trials in the pain rating task was increased to improve the signal-to-noise ratio. As shown in Fig. 4B, participants underwent two phases following pain calibration. In the rest phase,

participants focused on a fixation cross and remained still for 5 min, followed by 15 trials of the pain rating task. This process sequence was repeated twice, comprising the pre-task session. In the sensorimotor synchronization phase, participants engaged in one of the three stimulus conditions as in Experiment 2 for 5 min, followed by the same pain rating task as in the rest phase. This sequence was also repeated twice, forming the post-task session.

## EEG recording and preprocessing

In Experiment 3, EEG data were recorded using 32 Ag-AgCl scalp electrodes placed according to the international 10–20 system (ANT Neuro, the Netherlands) at a sampling rate of 1000 Hz. The CPz electrode served as the online reference, and all electrode impedances were kept below 10 kΩ. EEG preprocessing was conducted using EEGLAB[53], an open-source toolbox running under the MATLAB environment. Continuous EEG data were first filtered between 1 and 30 Hz, and then EEG epochs were extracted from –500 to 1000 ms relative to the onset of laser stimulus. Subsequently, the data were visually inspected to remove epochs contaminated by artifacts due to gross movements, and bad electrodes were interpolated using spherical spline interpolation. However, no electrodes required interpolation and only 1.26% of the epochs were discarded. Trials contaminated by eye blinks and movements were corrected using an independent component analysis algorithm (*runica* in EEGLAB).

## EEG time domain analysis

To reveal typical LEP waveforms, single-trial LEP waveforms for each participant were averaged separately for pre-task and post-task sessions. LEPs consist of multiple components that reflect distinct stages of cortical pain processing[43,54]. The N1 component, associated with early sensory-discriminative processing, typically exhibits a lateralized distribution that is contralateral to the stimulated hand[44,55]. The N2 and P2 components, linked to affective, cognitive, and attentional aspects of pain, show a robust vertex-centered distribution[56–58]. Based on their well-established functional distinctions and scalp distributions, the N1 component was measured at C3 (re-referenced to Fz) and the N2 and P2 components were measured at Cz (re-referenced to the average of the bilateral mastoids). For analysis, the N1 component was defined as the most negative deflection occurring between 160 and 220 ms after stimulus onset. The N2 and P2 components were identified and defined as the most negative deflection between 180 and 260 ms and the most positive deflection between 300 and 450 ms after stimulus onset, respectively. For each participant, peak latencies and local peak amplitudes (averaged within ±20 ms around each identified peak latency) were extracted. Subsequently, group-level LEP waveforms were obtained by averaging across participants for each session and each group, and group-level scalp topographies of N1, N2, and P2 waves were computed by spline interpolation at their respective peak latencies within the predefined time window.

## Statistics and reproducibility

To examine group differences in subjective ratings (i.e., pain intensity, unpleasantness, positive affect, and negative affect) and time-domain brain activity measures (i.e., the amplitude and latency of N1, N2, and P2 waves), we calculated the difference scores between the pre-task and post-task sessions (post minus pre). In addition, to assess participants' subjective perception of the task and to validate the effectiveness of the synchronization manipulation in Experiment 2, responses to the manipulation check questionnaire were analyzed. In Experiments 1–3, statistical analyses were conducted using two-way ANOVA (Experiment 1) or one-way ANOVA (Experiments 2 and 3) under a between-subjects design, when the assumption of normality was met. Otherwise, the Kruskal–Wallis $H$ test was applied as a non-parametric alternative. When significant main effects or interactions were found ($P < 0.05$), post hoc $t$ tests (parametric) or Mann–Whitney $U$-tests (non-parametric) were performed in two-sided, with Bonferroni correction applied to control for multiple comparisons. Partial eta-squared ($\eta_p^2$) was reported as the measure of effect size. Statistical analyses for the exploratory experiment,

which focused on resting-state EEG measures, are detailed in Supplementary Experiment.

To further explore relationships between pain ratings and self-reported measures from the manipulation check (Experiment 2), as well as between pain ratings and brain activities (Experiment 3), Spearman's rank correlation was conducted due to its robustness against outliers. In Experiment 3, mediation analyses were conducted to test whether the effects of perceived sensorimotor synchrony on pain ratings were mediated by stimulus-evoked brain responses. Since the independent variable (i.e., group) was categorical with three levels, dummy coding was applied, using the asynchrony group as the reference category. A percentile bootstrap estimation with 5000 bootstrapped samples was used to calculate bias-corrected 95% CIs of the indirect and direct effects. These effects were considered statistically significant at $P < 0.05$ in a two-sided test when the 95% CIs did not include zero[59].

## Reporting summary

Further information on research design is available in the Nature Portfolio Reporting Summary linked to this article.

## Data availability

The primary data are publicly available (https://osf.io/5z24n/)[60]. Additionally, the numerical source data underlying the graphs are also provided in an Excel file, which is designated as Supplementary Data 1.

## Code availability

The code for preprocessing and analysis is accessible in the repository hosted by the Open Science Foundation (https://osf.io/5z24n/)[60].

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

## Acknowledgements
X.L. is supported by the National Natural Science Foundation of China (32171077) and a grant from the Youth Innovation Promotion Association, CAS (2022084). L.H. is supported by the National Key Research and Development Program of China (2024YFC3607600) and Beijing Natural Science Foundation (JQ22018).

## Author contributions
X.P.: methodology, validation, formal analysis, investigation, data curation, visualization, writing—original draft, writing—review and editing; Y.X.: methodology, investigation, data curation; L.H.: resources, conceptualization, methodology, resources, writing—review and editing, supervision, funding acquisition; X.L.: conceptualization, methodology, formal analysis, validation, resources, writing—original draft, writing—review and editing, project administration, supervision, funding acquisition.

## Competing interests
The authors declare no competing interests.
