## [Transparent Peer Review file · Communications Biology]

Perceived sensorimotor synchrony enhances pain modulation and attenuates laser-evoked potentials

Corresponding Author: Dr Xuejing Lu

Version 0:

Reviewer comments:

Reviewer #1

(Remarks to the Author)

Thank you for the opportunity to review the manuscript titled "Inducing a state of flow through sensorimotor synchronization to modulate pain perception" Overall, the paper is well written, particularly the behavioral part. The investigation into EEG patterns associated with the experimental conditions represents an important line of research, as these findings have the potential to offer valuable insights into the mechanisms underlying pain modulation. However, there are several significant limitations in the current version of the study that need to be addressed.

- 1) The authors should provide justification and appropriate references for the choice of tempos at 90 BPM and 120 BPM used in the experiment.
- 2) LEPs: clarification is needed on why EEG data was epoched both before and after cleaning the data. Re-epoching may alter the noise characteristics in the data and potentially impact the resulting signals. Additionally, justification is needed for using only the Cz electrode as the electrode of interest. Since stimulation was applied to the hand, it would be more appropriate to examine LEPs across regions involved in primary and secondary pain processing, including frontal, central, and parietal electrodes.
- 3) Gamma power: the gamma power analysis should be extended to separately examine lower and higher gamma frequency bands, as this may yield more detailed insights into the underlying mechanisms of pain modulation.
- 4) Delta and theta power should also be analyzed, as they may offer additional information on the neurophysiological processes involved.
- 5) Figures:
 - Please provide the full PSD plot for Figure 6A before zooming in on the frequency bands of interest to ensure a complete view of power spectra. Please include delta and theta power.
 - Figures 3 through 6, in addition to labeling the x- and y-axes, include subplot headings to make the figures easier to follow.
- 6) Discussion:
 - "Importantly, the amplitude of the N2 component effectively mediated the reduction in pain ratings across the three conditions."

The interpretation of the N2 component appears unclear. Authors suggest a causal relationship that has not been substantiated. It is not clear whether changes in N2 amplitude are a result of reduced pain or a mediator of it. Furthermore, as previously mentioned, LEP data should be provided from a broader range of areas involved in pain processing.

 - Consider rephrasing section 4.2 subheading. "Sensorimotor synchronization plays an important role in modulating pain perception" to "Sensorimotor synchronization modulates pain perception".
 - "Our results suggest that in-phase sensorimotor synchronization may share similar mechanisms, promoting an immersive state that enhances pain modulation via increased γ -band activity".

To support this interpretation, a correlation analysis between changes in gamma-band power and pain ratings should be provided to demonstrate the relationship, while acknowledging that this does not imply causality.

 - "Therefore, the minimal reduction in α -band power observed in the occipital region after sensorimotor synchronization, relative to rest condition, may indicate visual disengagement, as visual information was not critical for the drumming to music task. This visual disengagement may further contribute to the overall analgesic effect by reducing competing sensory inputs".

To substantiate this claim, a correlation analysis between changes in alpha-band power and pain ratings is warranted.

Reviewer #2

(Remarks to the Author)

The manuscript "Inducing a state of flow through sensorimotor synchronization to modulate pain perception" presents four experiments on the effects of drumming to music on pain intensity and unpleasantness. The topic is timely and potentially of interest to readers of Communications Biology, and the manuscript is well-written and mostly clear. However, the experimental designs and selection (or omission) of dependent variables do not sufficiently support many of the conclusions presented in the discussion. Sensorimotor synchronization was not directly measured, there was no control condition with random movement, and immersion or flow, which are prominently mentioned, were not assessed. Additionally, the manuscript lacks clearly stated hypotheses, which makes it difficult to evaluate the theoretical and methodological grounding of the experiments. If the authors cannot address these issues, I recommend rejecting the manuscript.

First main issue:

We do not know whether participants successfully synchronized with the in-phase stimuli or how they synchronized with the anti-phase and asynchronous stimuli. It is possible that participants ignored the visual input and followed the auditory rhythm in the anti-phase or asynchronous conditions. It is also possible that synchronization accuracy and stability did not differ between in-phase, anti-phase, and asynchronous conditions, in which case the conclusion would be incorrect. The bottom line is: A study on the effects of sensorimotor synchronization must present measures of sensorimotor synchronization accuracy and stability.

Second main issue:

Without a random movement condition, it is hard to assess the contribution of sensorimotor synchronization compared to movement in general. This is an issue in all four experiments but especially critical in Experiment 4. As Experiment 4 only includes drumming in-phase conditions and a rest condition it is incorrect to attribute the effects to sensorimotor synchronization – the effect could as well be purely driven by physical activity versus rest.

Third main issue:

The manuscript focusses on flow and immersive states. Flow is even mentioned in the title. However, there is no assessment of these concepts and no empirical evidence indicating that flow was induced. In addition, statements like "Sensorimotor synchronization to music [...] induces a state of 'flow'" (abstract) are extremely overstated and overgeneralized. Occasionally, under specific circumstances, sensorimotor synchronization can play a part in inducing a state of flow. These kinds of overgeneralizations can be found in other places: "These processes not only enhance sensory integration and cognitive engagement, but also creates a state of 'flow'" (introduction); "Sensorimotor synchronization not only enhances cognitive engagement and attentional focus but also promotes a state of 'flow'" (discussion).

Fourth main issue:

The introduction fails to explain the theoretical and methodological background of the measures used and does not formulate hypotheses.

Other major issues:

EEG recording and preprocessing: "EEG epochs were extracted within -3000 to 2000 ms window" – time window relative to what? What is zero here? The segmentation is unclear. Later again: "extracted within -500 to 1000 ms time window" – here I assume that this is relative to the onset of the pain stimulation?

Experiment 3: Why were N2 and P2 used? Why the Cz electrode? This needs to be better explained.

Experiment 4: "five canonical frequency bands (δ : 1-3 Hz, θ : 4-7 Hz, α : 8-12 Hz, β : 13-29 Hz, and γ : 30-90 Hz)" – why those frequency bands?

"in Experiments 3 and 4, we determined electrode selection through multiple comparisons across all electrode sites" – this is unclear.

"Although the drumming groups also outperformed the nodrumming groups in reducing unpleasantness, the main effect of drumming did not reach statistical significance ($F(1,76) = 3.67$, $P = 0.059$, $\eta^2 = 0.05$)." – This statement is incorrect. There are no statistical trends or effects approaching significance. If the results is not significant then the drumming groups did NOT outperform the no-drumming groups.

"the more participants felt synchronized and relied on rhythmic auditory sequence, the greater their reductions in pain ratings" – Importantly, the felt synchrony and the actual synchrony (objectively measured) is not necessarily aligned. E.g., Matthews et al., 2022, Music Perception: "Perceived synchrony showed a stronger relation with groove ratings than measured synchrony". This is just another example of why objectively measured accuracy and stability of drumming need to

be reported.

From Figure 5B, I can roughly extract the mean N2 peaks, which are -12 for in-phase, -6 for anti-phase, and -7.5 for asynchrony. Why is the only reported pairwise comparison the significant difference between in-phase and asynchrony? Given that the peak of anti-phase is even lower (and the variance comparable) there should also be a significant in-phase versus anti-phase difference. This inconsistency has to be reported and explained. Leaving this comparison out is untransparent and cherry-picking.

What kind of filter was applied to Figure 6A top left? Shouldn't this be unfiltered EEG data?

Discussion: "pain relief was significantly enhanced when participants' drumming strokes were perfectly synchronized with the auditory drumbeats" – again: this statement does not follow from the data as sensorimotor synchronization was not measured. It is incorrect.

The whole discussion of Experiment 4 (Chapter 4.3) does not follow from the data and results. It could also be a pure effect of movement versus rest. With the current design it is impossible to make any claim about the specific effect of sensorimotor synchronization.

Minor issues:

Page numbers are missing.

Only the number of female participants is reported. This does not mean that the other participants were male. What about non-binary or other participants, or participants who would prefer not to provide this detail? Were these options provided?

The design of Experiment 1 should mention that it was a between-subject design with four groups.

The following sentences are unclear to me: "Given the pre-post experimental design, any group differences in the post-test, relative to the pre-test, cannot be attributed to attentional distraction during the intervention session. Instead, they reflect a more direct and uncontaminated measure of the aftereffects of different experimental manipulations."

Experiment 2, participants: "ANOVA focusing on fixed effects across three groups" – Here it should be mentioned whether this is a within or between design.

Experiment 3: "Group" is a confusing term for the within-subjects design. Maybe "condition"?

"drumming to music, a representative form of groove" – drumming to music can, under specific circumstances, induce an experience of groove, but drumming to music per se is not a representation of groove.

Reviewer #3

(Remarks to the Author)

This is a very impressive series of four experiments, with a total sample size of 244 participants. Understanding mechanisms underlying music induced analgesia is an important and timely topic. Not a lot of studies on sensorimotor synchronization but there are a few studies. The authors already cite Werner et al. (2023), but there is also another study that just came out that I think could be interesting to them (https://journals.lww.com/pain/fulltext/9900/individualizing_musical_tempo_to_spontaneous_rates.810.aspx).

In any case, this is a topic of growing relevance, and the current set of studies represents by far the most thorough and well-designed exploration to date. The findings convincingly demonstrate that sensorimotor synchronization yields stronger analgesic effects than either motor or auditory stimulation alone, and that in-phase synchronization is more effective than anti-phase or asynchronous conditions. The evidence pointing to the mediation of these effects by the N2 component of the ERP is particularly compelling—very nicely done.

That said, I do have some concerns about Study 4. This could be put in limitation, but study 4 could also be removed if other reviewers or editor feel the same way about it:

:

- The "rest" condition always occurs in the middle of the sequence and is not counterbalanced. This undermines any comparison involving the rest/silence condition and casts doubt on the conclusions drawn from Study 4.
- The sample size ($n = 20$) is quite small relative to the other experiments. Could the authors clarify how this sample size was determined?
- Why couldn't the frequency EEG analyses be performed on the participants from Study 3? This larger sample might have offered more statistical power.

A few minor suggestions:

- It would be helpful to briefly justify the inclusion of EEG measures in the introduction.
- I'm curious whether there were any differences between the 90 bpm and 120 bpm conditions. The recent study by Yi et al. (2025) suggests that 120 bpm may be more effective for most participants.

- It would be great to visualize the mediation pathway more clearly in the figure by including the $a*b$ path and indicating whether the mediation effect is statistically significant.
- One conceptual point worth at least acknowledging: is it possible that the superior effects of the sound + drumming and in-phase conditions reflect the fact that these are simply easier tasks? Synchronizing to an auditory cue may be more intuitive than to a visual one. If so, the observed effects might reflect task ease rather than sensorimotor coupling per se. Performance data (e.g., objective synchrony measures) could help clarify this. If there is a difference in performance, then perhaps that means that the effect is driven by task ease, not sensorimotor synchronization per se. For instance, we could predict to obtain similar results with chess puzzles of differing difficulties. While this is potentially important for the field in the long term, I think that this is more a question for another study. But if the authors agree, this is something that they could potentially discuss in the discussion section of the paper.

Version 1:

Reviewer comments:

Reviewer #1

(Remarks to the Author)

My comments have been appropriately addressed.
I thank the authors for their efforts.

Reviewer #2

(Remarks to the Author)

I am happy with the changes to the manuscript and want to thank the authors for their detailed and clear responses. I recommend accepting the manuscript for publication.

Reviewer #3

(Remarks to the Author)

I'm satisfied with the authors' thorough revisions and appreciated the other reviewers' thoughtful comments. Reviewer #2's first point regarding the absence of an objective measure of sensorimotor synchronization is particularly compelling. It's unfortunate this wasn't collected, as it would have been easy to do. This limitation affects Studies 2 and 3 more than Study 1, where only the music + drumming condition allows true synchronization. I also think that the drumming-only condition adequately controls for "random movement" there.

For Studies 2–3, the reviewer concern that participants may not have actually performed the task as intended is difficult to shake off. Self-reported reliance on visual cues is reassuring, but objective asynchrony measures would have been far more convincing.

Also, why not have reported these manipulation checks in the previous version of manuscript?

Overall, despite these imperfections, I think that the series of studies totalling more than 200 participants makes a meaningful contribution. However, we should keep in mind that that Study 1 primarily replicates Werner; the paper's novelty rests with Studies 2–3, which are the ones that really lack objective measures of asynchrony.

I'm genuinely torn: self-reports suggest participants followed the visual cue across conditions, yet I agree with Reviewer #2 that it doesn't make sense to study synchronization without directly measuring it. A practical compromise could be to test a small confirmatory sample ($n \approx 10-20$) with objective measures of asynchrony to verify that participants do attempt to follow the visual cues in all conditions. Although it would be ideal to find between-condition differences in asynchrony, I don't think that this is necessary. Even in the absence of differences, confirming task adherence would still support the intended interpretation. Of course, this approach assumes the new sample behaves similarly to the original cohorts, but I think that it would be a reasonable assumption to make here.

On last minor comments: can the authors replace their "delta" sign with "post-pre" if this is the difference that they tested. It is hard to follow the signs, especially in the mediation graph. The $N2$ is a negative component presented upward, and then a difference score is extracted from that negative component, it is easy to lose track of the signs. Please confirm that path a is in the expected direction, i.e. a positive coefficient indicates that the synchronous condition is associated with a greater reduction in $N2$ (or less negative $N2$).

Reviewers' comments:

Reviewer #1 (Remarks to the Author):

Thank you for the opportunity to review the manuscript titled “Inducing a state of flow through sensorimotor synchronization to modulate pain perception” Overall, the paper is well written, particularly the behavioral part. The investigation into EEG patterns associated with the experimental conditions represents an important line of research, as these findings have the potential to offer valuable insights into the mechanisms underlying pain modulation. However, there are several significant limitations in the current version of the study that need to be addressed.

Reply: We thank the reviewer for the positive feedback on our manuscript. A point-by-point response to the comments follows.

1. The authors should provide justification and appropriate references for the choice of tempos at 90 BPM and 120 BPM used in the experiment.

Reply: The tempos of 90 bpm and 120 bpm were chosen because they fall within the commonly reported optimal tempo range for human sensorimotor synchronization. Prior studies have shown that rhythmic movements are most naturally and accurately synchronized within this range, facilitating entrainment with external stimuli (e.g., Repp & Su, 2013; van Noorden & Moelants, 1999). Moreover, a recent study by Yi et al. (2025) further supports the importance of aligning musical tempo with individuals' spontaneous production rates (SPRs) to enhance analgesic effects. Based on these theoretical and empirical considerations, we believe that our tempo choices were appropriate for facilitating effective sensorimotor synchronization while also being relevant for investigating rhythm-based pain modulation.

We have now added this justification and supporting references in the revised manuscript (see lines 124-127).

References:

Repp, B. H. & Su, Y. H. Sensorimotor synchronization: a review of recent research (2006-2012). *Psychon. Bull. Rev.* **20** (2013).

Van Noorden, L., & Moelants, D. Resonance in the perception of musical pulse. *J. New Music Res.* **28**, 43-66 (1999).

Yi, W., Palmer, C., Serian, A., & Roy, M. Individualizing musical tempo to spontaneous rates maximizes music-induced hypoalgesia. *Pain* **10**, 1097 (2022).

2. LEPs: clarification is needed on why EEG data was epoched both before and after cleaning the data. Re-epoching may alter the noise characteristics in the data and potentially impact the resulting signals.

Reply: We apologize for the imprecise wording in the original manuscript. The term "further extracted" referred to the selection of a shorter time window from the initially defined longer epochs for better LEP feature extraction and visualization. To improve data quality during preprocessing, we initially segmented the EEG into relatively long epochs (–3 to 2 s relative to pain stimulus onset) before ICA. This approach allowed us to preserve more data points, especially in the pre-stimulus interval, and improve the reliability of artifact component identification. Subsequently, a shorter analysis window (–0.5 to 1 s relative to pain stimulus onset) was selected from these epochs for LEP feature extraction and visualization.

Nevertheless, to ensure that this procedure did not introduce any unintended bias or alter the noise characteristics of the signal, we reconducted the analysis using data epoched directly within the final analysis window (–0.5 to 1 s relative to pain stimulus onset) prior to ICA. The new results were highly consistent with those reported in the original manuscript.

We have revised the Methods section accordingly (see lines 273-277) and updated the Results section in the revised manuscript with these new analyses (see lines 430-437 and 452-467).

3. Additionally, justification is needed for using only the Cz electrode as the electrode of interest. Since stimulation was applied to the hand, it would be more appropriate to examine LEPs

across regions involved in primary and secondary pain processing, including frontal, central, and parietal electrodes.

Reply: LEPs consist of multiple components that reflect distinct stages of cortical pain processing (e.g., Garcia-Larrea et al., 2003; Iannetti et al., 2005). The early N1 component (~160-220 ms post-stimulus), associated with the activity in the primary somatosensory cortex, typically shows a lateralized distribution contralateral to the stimulated site. This component is thought to reflect early sensory-discriminative processing of nociceptive input. This component is usually analyzed at contralateral central electrodes with Fz as the reference (i.e., C3-Fz for right hand stimulation and C4-Fz for left hand stimulation; e.g., Hu et al., 2010; Valentini et al., 2012).

The later N2 and P2 components (~180-450 ms) exhibit a robust vertex-centered distribution and are considered to originate from the anterior cingulate cortex and bilateral insula. These components are linked to affective, cognitive, and attentional aspects of pain. For this reason, Cz is widely used as the site of interest for analyzing the N2/P2 complex in almost all LEP studies (e.g., Iannetti et al., 2008; Tiemann et al., 2018; Zhang et al., 2022).

Based on these well-established functional distinctions and spatial distributions, we focused our primary analyses of the N2 and P2 components at Cz. In response to the reviewer's suggestion, we have now also included an analysis of the N1 component at the central electrode contralateral to the stimulated hand (i.e., C3-Fz; Figure R1).

These expanded justifications and analyses are fully integrated into the revised manuscript. The detailed explanation of LEP components and electrode selection can be found in the Methods section (see lines 290-292), and the corresponding N1 results have been added to the Results section (see lines 436-437) and Supplementary Information (see Figure S2).

Figure R1. Average waveforms at C3 electrode (with Fz as the reference electrode) of each group in response to nociceptive laser stimuli in the pre-task (solid line) and the post-task (dash line) sessions (black lines were difference waves between two sessions). No significant group differences were observed in changes of N1 amplitude and latency.

References:

Garcia-Larrea, L., Frot, M. & Valeriani, M. Brain generators of laser-evoked potentials: from dipoles to functional significance. *Neurophysiol. Clin.* **33**, 279-29 (2003).

Crucchi, G. *et al.* Recommendations for the clinical use of somatosensory-evoked potentials. *Clin. Neurophysiol.* **119**, 1705–1719 (2008).

Hu, L., & Iannetti, G. D. Neural indicators of perceptual variability of pain across species. *Proc. Natl. Acad. Sci. U. S. A.* **116**, 1782–1791 (2019).

Hu, L., Mouraux, A., Hu, Y., & Iannetti, G. D. A novel approach for enhancing the signal-to-noise ratio and detecting automatically event-related potentials (ERPs) in single trials. *Neuroimage* **50**, 99-111 (2010).

Iannetti, G. D., Hughes, N. P., Lee, M. C., & Mouraux, A. Determinants of laser-evoked EEG responses: pain perception or stimulus saliency? *J. Neurophysiol.* **100**, 815-828 (2008).

Iannetti, G. D., Zambreanu, L., Crucchi, G. & Tracey, I. Operculoinsular cortex encodes pain intensity at the earliest stages of cortical processing as indicated by amplitude of laser-evoked potentials in humans. *Neuroscience* **131**, 199-208 (2005).

Tiemann, L., Hohn, V. D., Ta Dinh, S., May, E. S., Nickel, M. M., Gross, J., & Ploner, M. Distinct

patterns of brain activity mediate perceptual and motor and autonomic responses to noxious stimuli. *Nat. Commun.* **9**, 4487 (2018).

Valentini, E., Hu, L., Chakrabarti, B., Hu, Y., Aglioti, S. M., & Iannetti, G. D. The primary somatosensory cortex largely contributes to the early part of the cortical response elicited by nociceptive stimuli. *Neuroimage* **59**, 1571-1581 (2012).

Zhang, L. B., Lu, X. J., Huang, G., Zhang, H. J., Tu, Y. H., Kong, Y. Z., & Hu, L. Selective and replicable neuroimaging-based indicators of pain discriminability. *Cell Rep. Med.* **3(12)**, 100846 (2022).

4. Gamma power: the gamma power analysis should be extended to separately examine lower and higher gamma frequency bands, as this may yield more detailed insights into the underlying mechanisms of pain modulation.

Reply: As recommended, we conducted a more fine-grained analysis of the γ -band by separating it into low γ (30–60 Hz) and high γ (61–90 Hz) sub-bands, following common practices in EEG studies (e.g., Li et al., 2023; Linde et al., 2023). This refined analysis, which is part of Experiment 4 and now fully presented in Supplementary Experiment, revealed that the significant changes in γ -band power reported in the original manuscript were predominantly driven by effects within the high γ -band. While we observed some isolated electrodes with significant effects in the low γ -band, no significant clusters of spatially adjacent electrodes were identified. In contrast, the high γ -band exhibited robust and consistent effects that align with our initial findings (see Figure R2).

These comprehensive updated findings, including specific statistical details and scalp topographies for both low and high γ -bands, are now fully incorporated into the Results section of the Supplementary Experiment (see Supplementary Information, lines 90-101 and Figures S4 & S5).

Figure R2. The scalp topographies of low and high γ -band oscillations in the pre-task and post-task sessions for the rest, 90-bpm, and 120-bpm conditions. Significant effects of condition on changes in PSD were broadly observed within the high γ -band, particularly electrodes over the centroparieto-occipital area (marked with white dots). In contrast, while some isolated electrodes showed significant effects in the low γ -band, no significant clusters of spatially adjacent electrodes were identified.

References:

Li Z, Zhang L, Zeng Y, Zhao Q, Hu L. Gamma-band oscillations of pain and nociception: A systematic review and meta-analysis of human and rodent studies. *Neurosci. Biobehav. Rev.* **146**, 105062 (2023).

Linde, L. D., Ortiz, O., Choles, C. M., & Kramer, J. L. K. Pain-related gamma band activity is dependent on the features of nociceptive stimuli: a comparison of laser and contact heat. *J Neurophysiol.* **129**, 262–270 (2023).

5. Delta and theta power should also be analyzed, as they may offer additional information on the neurophysiological processes involved.

Reply: In response, we have conducted PSD analyses for both δ and θ bands, as well as β -band, along with other frequency bands. However, these analyses did not reveal any statistically

significant differences among conditions (see Figure R3).

For the sake of completeness and transparency, the results of all investigated frequency bands, including δ , θ , and β , have now been incorporated into the Supplementary Experiment. The corresponding scalp topographies are also presented (see Supplementary Information, lines 100-101 and Figure S5).

Figure R3. The scalp topographies of δ -, θ -, and β -band oscillations in the pre-task and post-task sessions for the rest, 90-bpm, and 120-bpm conditions. While significant effect of condition was observed at CP6 electrode within the β -band (marked with white dots), no significant clusters of spatially adjacent electrodes were identified for all conditions. Pre.: the pre-rsEEG session; Post.: the post-rsEEG session; PSD: power spectral density.

6. Figures:

a) Please provide the full PSD plot for Figure 6A before zooming in on the frequency bands of interest to ensure a complete view of power spectra. Please include delta and theta power.

Reply: As recommended, we have replaced the original Figure 6A with fully revised Figure S4 in Supplementary Information. This updated figure now presents the complete PSD plot across the entirely investigated frequency range (1-90 Hz; see Figure R4).

Figure R4. Averaged power spectral density (PSD) of spontaneous EEG oscillations at Pz electrode in the pre-task (solid line) and post-task (dash line) sessions for the rest (green), 90-bpm (blue), and 120-bpm (red) conditions.

b) Figures 3 through 6, in addition to labeling the x- and y-axes, include subplot headings to make the figures easier to follow.

Reply: We have now added descriptive subplot headings to all relevant figures, including original Figures 3-6, now corresponding to Figures 3, 4, 5 in the main manuscript and Figures S2, S4, S5 in the Supplementary Information).

Discussion:

7. *“Importantly, the amplitude of the N2 component effectively mediated the reduction in pain ratings across the three conditions.”*

The interpretation of the N2 component appears unclear. Authors suggest a causal relationship that has not been substantiated. It is not clear whether changes in N2 amplitude are a result of reduced pain or a mediator of it. Furthermore, as previously mentioned, LEP data should be provided from a broader range of areas involved in pain processing.

Reply: We thank the reviewer for this important comment. We fully acknowledge that mediation analysis, especially in non-randomized designs, cannot establish causal relationships. In our case, the proposed mediator (i.e., changes in N2 amplitude) is an EEG-derived measure, which was not experimentally manipulated. It is therefore possible that unmeasured or uncontrolled confounding variables may have simultaneously influenced both the neural response (changes in N2 amplitude) and the behavioral outcome (reductions in pain ratings), potentially leading to a spurious association.

Nevertheless, our intention with the mediation analysis was to identify neurophysiological correlates that statistically account for a portion of the observed behavioral pain modulation, providing evidence for the underlying mechanisms rather than asserting a direct causal influence from N2 changes to pain reduction.

To address this concern and avoid overinterpretation, we have revised the manuscript to describe the relationship in purely correlational and associational terms, and provided a more precise and cautious interpretation of N2 amplitude changes as a neurophysiological indicator that statistically accounts for part of the observed pain modulation, rather than a definitive mediator in a causal sense (see lines 476-478 and 548-550).

8. Consider rephrasing section 4.2 subheading. “Sensorimotor synchronization plays an important role in modulating pain perception” to “Sensorimotor synchronization modulates pain perception”.

Reply: We have revised the subheading of Section 4.2 to “*Perceived sensorimotor synchrony modulates pain perception*” as recommended (see line 518).

9. *“Our results suggest that in-phase sensorimotor synchronization may share similar mechanisms, promoting an immersive state that enhances pain modulation via increased γ -band activity”. To support this interpretation, a correlation analysis between changes in gamma-band power and pain ratings should be provided to demonstrate the relationship, while acknowledging that this does not imply causality.*

Reply: We thank the reviewer for this valuable suggestion. Our initial interpretation, linking in-phase sensorimotor synchronization to an immersive state and enhanced pain modulation via increased γ -band activity, was indeed primarily based on established prior findings. Specifically, this interpretation drew from literature reporting a significant negative correlation between spontaneous γ -band activity and subsequent pain ratings in immersive contexts, where increased γ -band power was associated with enhanced engagement and reduced pain perception (see Li et al., 2023).

However, we fully acknowledge the critical limitation that Experiment 4 was not designed to collect trial-by-trial pain ratings. Consequently, we are unable to directly assess the relationship between γ -band power and subjective pain experience in this dataset. In light of this limitation, and after careful consideration of the feedback from other reviewers regarding Experiment 4's broader design constraints, the entire content of Experiment 4 has been moved to Supplementary Information and is now presented as an exploratory Supplementary Experiment. Within this Supplementary Experiment, we explicitly and extensively discuss its inherent design limitations, including its inability to fully disentangle synchrony-specific effects from general rhythmic movement alone, and critically, the absence of trial-by-trial pain ratings.

Accordingly, we have revised the relevant discussion within Supplementary Experiment. Our language now downplays any definitive interpretation, clarifying that our hypothesis regarding immersion and γ -band activity remains speculative and is primarily grounded in prior literature, rather than being directly supported by our current data (see Supplementary Information, lines 130-135, and 145-156).

Reference:

Li, J. *et al.* The analgesic effects and neural oscillatory mechanisms of virtual reality scenes based

on distraction and mindfulness strategies in human volunteers. *Br. J. Anaesth.* **131**, 1082-1092 (2023).

10. "Therefore, the minimal reduction in α -band power observed in the occipital region after sensorimotor synchronization, relative to rest condition, may indicate visual disengagement, as visual information was not critical for the drumming to music task. This visual disengagement may further contribute to the overall analgesic effect by reducing competing sensory inputs". To substantiate this claim, a correlation analysis between changes in alpha-band power and pain ratings is warranted.

Reply: Similar to our reply to the previous comment, the original statement regarding α -band activity was intended as a speculative interpretation grounded in prior literature, which has linked reduced occipital α -band activity to increased visual attention and, conversely, enhanced α -band activity to visual disengagement (e.g., Klimesch, 2012; Jensen et al., 2002).

As detailed in our response to the previous comment, Experiment 4 (now designated as a Supplementary Experiment) did not include trial-by-trial pain ratings, which prevents us from performing a direct correlation analysis between changes in α -band power and pain perception within this dataset.

Accordingly, we have revised the corresponding discussion to reflect a more cautious interpretation. Rather than implying a direct modulatory relationship, we have now described the observed stabilization of occipital α -band activity as possibly consistent with a disengagement of visual attention, which might contribute to multisensory resource reallocation based on previous findings (e.g., Iurilli et al., 2012). This revision avoids causal language and clarifies that our interpretation remains speculative (see Supplementary Information, lines 139-144 and 152-156).

References:

Klimesch, W. Alpha-band oscillations, attention, and controlled access to stored information. *Trends Cogn. Sci.* **16**, 606-617 (2012).

Jensen, O., Gelfand, J., Kounios, J. & Lisman, J. E. Oscillations in the alpha band (9-12 Hz) increase

with memory load during retention in a short-term memory task. *Cereb. Cortex* **12**, 877-882 (2002).

Iurilli, G. *et al.* Sound-driven synaptic inhibition in primary visual cortex. *Neuron* **73**, 814-828 (2012).

Reviewer #2 (Remarks to the Author):

The manuscript “Inducing a state of flow through sensorimotor synchronization to modulate pain perception” presents four experiments on the effects of drumming to music on pain intensity and unpleasantness. The topic is timely and potentially of interest to readers of Communications Biology, and the manuscript is well-written and mostly clear. However, the experimental designs and selection (or omission) of dependent variables do not sufficiently support many of the conclusions presented in the discussion. Sensorimotor synchronization was not directly measured, there was no control condition with random movement, and immersion or flow, which are prominently mentioned, were not assessed. Additionally, the manuscript lacks clearly stated hypotheses, which makes it difficult to evaluate the theoretical and methodological grounding of the experiments. If the authors cannot address these issues, I recommend rejecting the manuscript.

Reply: We sincerely thank the reviewer for the positive comments and constructive feedback. We fully acknowledge the concerns regarding the mismatch between some of the conclusions and the experimental designs. In response, we have made substantial revisions to the manuscript to address these important issues. Detailed point-by-point responses to each concern are provided below.

1. First main issue:

We do not know whether participants successfully synchronized with the in-phase stimuli or how they synchronized with the anti-phase and asynchronous stimuli. It is possible that participants ignored the visual input and followed the auditory rhythm in the anti-phase or asynchronous conditions. It is also possible that synchronization accuracy and stability did not differ between in-phase, anti-phase, and asynchronous conditions, in which case the conclusion would be incorrect. The bottom line is: A study on the effects of sensorimotor synchronization must present measures of sensorimotor synchronization accuracy and stability.

Reply: We agree with the reviewer that the absence of objective measures of sensorimotor synchronization (e.g., mean asynchrony, standard deviation of asynchrony) limits the

interpretability of our findings. As noted, it is possible that participants synchronized similarly across the in-phase, anti-phase, and asynchronous conditions, or that they may have ignored the visual input and simply followed the auditory rhythm.

In our original design, we relied on participants' subjective reports of perceived synchrony to infer sensorimotor alignment. The core of this study lies in perceived synchrony. Our experimental design aimed to induce varying subjective experiences of temporal alignment between participants' movements and external rhythms by systematically manipulating the alignment between auditory and visual cues, rather than directly measuring objective temporal alignment accuracy. In essence, we indirectly modulated participants' sensorimotor synchronization with the auditory rhythm by varying the audio-visual congruency of the external stimuli. To validate the effectiveness of this perceived synchrony manipulation, we conducted a detailed manipulation check in Experiment 2. The results clearly demonstrated the following and showed in Figure R5:

- (1) Perceived synchrony differed significantly across conditions ($H_{(2)} = 34.95, P < 0.001$). Specifically, the in-phase synchrony group reported the strongest perceived synchrony, significantly higher than both the anti-phase and asynchrony groups. This suggests that our manipulation successfully induced distinct levels of subjective synchronous experience.
- (2) Addressing the concern that participants might have ignored visual input and simply followed the auditory rhythm, our further analyses revealed that no significant group differences were observed in reported reliance on visual cues ($H_{(2)} = 3.11, P = 0.21$). This indicates that participants across all groups attempted to follow the visual flashing-dot cues as instructed. However, for reliance on auditory cues, the in-phase synchrony group reported significantly higher reliance ($H_{(2)} = 20.60, P < 0.001$). In other words, in the in-phase condition, where visual and auditory cues were highly congruent, the prominence of auditory cues was amplified, further enhancing their perceived synchrony, while in the anti-phase and asynchrony conditions, the audio-visual mismatch likely led to lower reliance on auditory cues and a weaker perceived synchrony.

Figure R5. Participants' ratings of perceived audio-visual synchrony (left), auditory-cue reliance (middle), and visual-cue reliance (right) across three conditions: in-phase synchrony (green), anti-phase synchrony (blue), and asynchrony (red). Error bar represents \pm one standard error of the mean; *** $P < 0.001$; ** $P < 0.01$; * $P < 0.05$.

Importantly, our study establishes a direct link between perceived synchrony and pain reduction. Empirical data show that changes in pain ratings (including pain intensity and unpleasantness) were significantly negatively correlated with participants' perceived synchrony (see Figure R6; pain intensity: $\rho = -0.53$, $P < 0.001$; unpleasantness: $\rho = -0.27$, $P = 0.028$). Furthermore, changes in pain intensity were also significantly negatively correlated with the extent to which participants relied on auditory drumbeats (see Figure R6; pain intensity: $\rho = -0.39$, $P = 0.001$; unpleasantness: $\rho = -0.25$, $P = 0.040$). These robust correlations demonstrate that the subjective experience of alignment, and the active engagement with the rhythm, are key factors of the analgesic effect. This aligns with findings from Matthews et al. (2022), which showed that perceived motor synchrony with the beat is more strongly related to groove than measured synchrony.

Figure R6. Correlations between changes in pain ratings (both pain intensity and unpleasantness) and

the degree to which participants felt the synchronization as well as the extent to which they relied on the drumbeats for drumming in Experiment 2. Error bar represents \pm one standard error of the mean; *** $P < 0.001$; ** $P < 0.01$; * $P < 0.05$.

Nevertheless, we acknowledge that objective synchrony measures would provide more comprehensive context and deeper mechanistic insights into our findings. By focusing on perceived synchrony, this study has revealed its critical role in pain modulation. However, the complex interaction between objective and subjective synchrony, and how they individually or synergistically influence pain experience, represents an important topic for future research.

In the revised manuscript, we have explicitly stated that this study primarily focused on perceived synchrony rather than objectively quantified synchrony (see lines 40-57 and 71-77). As suggested, we have also acknowledged the absence of objective synchronization measures as a methodological limitation and highlighted its significance as an important direction for future research (see lines 566-573). Additionally, the detailed analysis and results of the manipulation check in Experiment 2 have been added to the manuscript to further validate the effectiveness of our experimental manipulation (see lines 371-389, and Figure S1).

Reference:

Matthews, T. E., Witek, M. A., Thibodeau, J. L., Vuust, P. & Penhune, V. B. Perceived motor synchrony with the beat is more strongly related to groove than measured synchrony. *Music Percept.* **39**, 423-442 (2022).

2. Second main issue:

Without a random movement condition, it is hard to assess the contribution of sensorimotor synchronization compared to movement in general. This is an issue in all four experiments but especially critical in Experiment 4. As Experiment 4 only includes drumming in-phase conditions and a rest condition it is incorrect to attribute the effects to sensorimotor synchronization – the effect could as well be purely driven by physical activity versus rest.

Reply: We fully agree that distinguishing the effects of sensorimotor synchronization from

those of movement *per se* is critical to isolating the underlying mechanism. In our prior research (Lu et al., 2021), we found that rhythmic physical activity alone (swinging hands at ~5 Hz for 5 seconds) can induce short-term analgesic effects, indicating that movement itself can contribute to pain modulation.

Our experimental design across Experiments 1-3 was specifically crafted to address this distinction within the main manuscript. To address this, Experiment 1 included a drumming-only group as a movement baseline. The results showed that participants in the two listening groups experienced greater reductions in pain intensity than those in the two no-listening groups, regardless of whether they engaged in drumming (drumming-only vs. drumming-and-listening: $P < 0.001$) or not (silence vs. listening-only: $P = 0.023$), supporting the idea that synchronization with rhythmic auditory input enhances pain reduction beyond movement alone (see Figure 3).

Furthermore, in Experiments 2 and 3, we included anti-phase and asynchronous conditions that involved movement but lacked temporal alignment with auditory input. These stimulus conditions served as non-synchronous movement controls. Our findings consistently showed that in-phase synchrony yielded significantly greater reductions in both subjective pain ratings and laser-evoked potentials compared to non-synchronous conditions (see Figures 4A and 5A). This further supports the notion that perceived synchrony *per se*, rather than movement alone, plays a critical role in modulating pain.

We acknowledge the reviewer's specific concern regarding Experiment 4. This experiment, which compared in-phase drumming to a rest condition, was primarily designed as an exploratory investigation into spontaneous neural oscillations, with the rest condition chosen to provide a stable neural baseline. In light of its design limitations, in careful consideration of the feedback from the other reviewers, the entire content of Experiment 4 has been moved to Supplementary Information.

We now explicitly discuss this experiment's inability to fully disentangle the effects of synchrony from those of general rhythmic movement alone, and how these factors influence the interpretability of its findings. Future studies should indeed include non-synchronous

movement control conditions in the context of spontaneous brain oscillations to further dissociate the effects of movement and sensorimotor synchronization.

These changes have been made in both the main manuscript (see lines 573-579) and Supplementary Experiment (see Supplementary Information, lines 145-149).

Reference:

Lu, X., Yao, X., Thompson, W. F., & Hu, L. Movement-induced hypoalgesia: behavioral characteristics and neural mechanisms. *Ann. N. Y. Acad. Sci.* **1497**, 39-56 (2021).

3. Third main issue:

The manuscript focusses on flow and immersive states. Flow is even mentioned in the title. However, there is no assessment of these concepts and no empirical evidence indicating that flow was induced. In addition, statements like “Sensorimotor synchronization to music [...] induces a state of ‘flow’” (abstract) are extremely overstated and overgeneralized. Occasionally, under specific circumstances, sensorimotor synchronization can play a part in inducing a state of flow. These kinds of overgeneralizations can be found in other places: “These processes not only enhance sensory integration and cognitive engagement, but also creates a state of ‘flow’” (introduction); “Sensorimotor synchronization not only enhances cognitive engagement and attentional focus but also promotes a state of ‘flow’” (discussion).

Reply: We acknowledge that our original manuscript may have overstated the association between sensorimotor synchronization and the induction of a flow state, especially in the absence of direct empirical measures such as self-report scales or behavioral measures. In our revision, we have carefully removed or rephrased statements that may have overstated the link between sensorimotor synchronization and flow.

This moderation of language is further reinforced by our decision to move Experiment 4, which explored exploratory neurophysiological insights into spontaneous brain oscillations potentially related to immersive engagement, to Supplementary Information. This ensures that the main manuscript exclusively focuses on directly measured behavioral and laser-evoked

potential data, thereby avoiding any potentially overstated inferences regarding flow or immersive states without direct subjective assessment.

We now clearly highlight that while sensorimotor synchronization can contribute to flow under specific circumstances (Stupacher, 2019), the present study did not directly assess flow and therefore cannot make definitive claims about its induction. Furthermore, we have modified the title to avoid overemphasizing the concept of flow (see main manuscript, lines 503-508, 570-573, and 579-581; Supplementary Information, lines 149-156).

Reference:

Stupacher, J. The experience of flow during sensorimotor synchronization to musical rhythms. *Music Sci.* **23**, 348-361 (2019).

4. Fourth main issue:

The introduction fails to explain the theoretical and methodological background of the measures used and does not formulate hypotheses.

Reply: As suggested, we have substantially revised the Introduction to more clearly explain the theoretical and methodological background of the key measures employed in our study (see lines 81-86). Specifically, we have now explained why pain ratings and LEPs were chosen, and how these measures are theoretically linked to pain modulation, cognitive engagement, and sensorimotor synchronization. Additionally, we have added explicit hypotheses for each of the experiments, grounded in prior research (see lines 77-87).

Other major issues:

5. EEG recording and preprocessing: “EEG epochs were extracted within -3000 to 2000 ms window” – time window relative to what? What is zero here? The segmentation is unclear. Later again: “extracted within -500 to 1000 ms time window” – here I assume that this is relative to the onset of the pain stimulation?

Reply: The zero time point in all epoching procedures corresponds to the onset of nociceptive laser stimuli. To improve data quality during preprocessing, we initially segmented the EEG into relatively long epochs (–3 to 2 s relative to pain stimulus onset) before ICA. This approach allowed us to preserve more data points, especially in the pre-stimulus interval, and improve the reliability of artifact component identification. Subsequently, a shorter analysis window (–0.5 to 1 s relative to pain stimulus onset) was selected from these epochs for LEP feature extraction and visualization. As suggested by another reviewer, this two-step procedure may introduce bias or alter noise characteristics. Therefore, we reconducted the analysis using data epoched directly within the final analysis window (–0.5 to 1 s relative to pain stimulus onset) prior to ICA. The new results were highly consistent with those reported in the original manuscript.

We have revised the Methods section accordingly (see lines 273-275) and updated the Results section in the revised manuscript with these new analyses (see lines 430-437 and 452-467).

6. Experiment 3: Why were N2 and P2 used? Why the Cz electrode? This needs to be better explained.

Reply: LEPs consist of multiple components that reflect distinct stages of cortical pain processing (e.g., Garcia-Larrea et al., 2003; Iannetti et al., 2005). The early N1 component (~160-220 ms post-stimulus), associated with the activity in the primary somatosensory cortex, typically shows a lateralized distribution contralateral to the stimulated site. This component is thought to reflect early sensory-discriminative processing of nociceptive input. This component is usually analyzed at contralateral central electrodes with Fz as the reference (i.e., C3-Fz for right hand stimulation and C4-Fz for left hand stimulation; e.g., Hu et al., 2010; Valentini et al., 2012).

The later N2 and P2 components (~180-450 ms) exhibit a robust vertex-centered distribution and are considered to originate from the anterior cingulate cortex and bilateral insula. These components are linked to affective, cognitive, and attentional aspects of pain. For this reason, Cz is widely used as the site of interest for analyzing the N2/P2 complex in almost all LEP

studies (e.g., Iannetti et al., 2008; Tiemann et al., 2018; Zhang et al., 2022).

Based on this well-established functional and spatial dissociation, we focused our primary analyses of the N2 and P2 components at Cz. In response to the reviewer's suggestion, we have now also included an analysis of the N1 component at the central electrode contralateral to the stimulated hand (i.e., C3-Fz).

These expanded justifications and analyses are fully integrated into the revised manuscript. The detailed explanation of LEP components and electrode selection can be found in the Methods section (see lines 283-292), and the corresponding N1 results have been added to the Results section (see lines 436-437) and Supplementary Information (see Figure S2).

References:

Garcia-Larrea, L., Frot, M. & Valeriani, M. Brain generators of laser-evoked potentials: from dipoles to functional significance. *Neurophysiol. Clin.* **33**, 279-29 (2003).

Crucchi, G. *et al.* Recommendations for the clinical use of somatosensory-evoked potentials. *Clin. Neurophysiol.* **119**, 1705–1719 (2008).

Hu, L., & Iannetti, G. D. Neural indicators of perceptual variability of pain across species. *Proc. Natl. Acad. Sci. U. S. A.* **116**, 1782–1791 (2019).

Hu, L., Mouraux, A., Hu, Y., & Iannetti, G. D. A novel approach for enhancing the signal-to-noise ratio and detecting automatically event-related potentials (ERPs) in single trials. *Neuroimage* **50**, 99-111 (2010).

Iannetti, G. D., Hughes, N. P., Lee, M. C., & Mouraux, A. Determinants of laser-evoked EEG responses: pain perception or stimulus saliency? *J. Neurophysiol.* **100**, 815-828 (2008).

Iannetti, G. D., Zambreanu, L., Crucchi, G. & Tracey, I. Operculoinsular cortex encodes pain intensity at the earliest stages of cortical processing as indicated by amplitude of laser-evoked potentials in humans. *Neuroscience* **131**, 199-208 (2005).

Tiemann, L., Hohn, V. D., Ta Dinh, S., May, E. S., Nickel, M. M., Gross, J., & Ploner, M. Distinct patterns of brain activity mediate perceptual and motor and autonomic responses to noxious stimuli.

Nat. Commun. **9**, 4487 (2018).

Valentini, E., Hu, L., Chakrabarti, B., Hu, Y., Aglioti, S. M., & Iannetti, G. D. The primary somatosensory cortex largely contributes to the early part of the cortical response elicited by nociceptive stimuli. *Neuroimage* **59**, 1571-1581 (2012).

Zhang, L. B., Lu, X. J., Huang, G., Zhang, H. J., Tu, Y. H., Kong, Y. Z., & Hu, L. Selective and replicable neuroimaging-based indicators of pain discriminability. *Cell Rep. Med.* **3(12)**, 100846 (2022).

7. Experiment 4: “five canonical frequency bands (δ : 1-3 Hz, θ : 4-7 Hz, α : 8-12 Hz, β : 13-29 Hz, and γ : 30-90 Hz)” – why those frequency bands?

Reply: As detailed in Experiment 4 (now designated as a Supplementary Experiment), this exploratory experiment was to investigate the effects of drumming and listening on resting-state EEG, which are not yet well established. Therefore, we analyzed the commonly used canonical frequency bands. This choice follows the recommendations and publication guideline for frequency domain analysis of EEG data (e.g., Keil et al., 2022). These bands are widely recognized in EEG research and correspond to distinct neurophysiological processes related to attention, arousal, sensory integration, and cognitive engagement. We deliberately excluded frequency ranges below 1 Hz and above 90 Hz, as they are more prone to slow drifts and high-frequency muscle artifacts, respectively. While minor variations in the exact boundaries may exist across studies, the frequency ranges we used represent a widely accepted and typical classification in EEG research (e.g., Başar et al., 2001; Buzsáki & Draguhn, 2004).

We have now added these justifications and references in the Methods section of Supplementary Experiment (see Supplementary Information, line 63).

References:

Başar, E., Başar-Eroglu, C., Karakaş, S., & Schürmann, M. Gamma, alpha, delta, and theta oscillations govern cognitive processes. *Int. J. Psychophysiol.* **39**, 241-248 (2001).

Buzsáki, G., & Draguhn, A. Neuronal oscillations in cortical networks. *Science* **304**, 1926-1929 (2004).

Keil, A., Bernat, E. M., Cohen, M. X., Ding, M., Fabiani, M., Gratton, G., ... & Weisz, N. Recommendations and publication guidelines for studies using frequency domain and time-frequency domain analyses of neural time series. *Psychophysiology* **59**, e14052 (2022).

8. “in Experiments 3 and 4, we determined electrode selection through multiple comparisons across all electrode sites” – this is unclear.

Reply: We thank the reviewer for highlighting the ambiguity in this statement. We apologize for the imprecision in the original manuscript. For Experiment 3, our approach to electrode selection for LEPs was not based on multiple comparisons across all electrode sites. Instead, the analysis of specific LEP components (N1, N2, and P2) was performed on pre-defined, theoretically driven representative electrodes (C3 and Cz). As explained in detail in our reply to Comment #6, this selection is based on well-established literature regarding the typical scalp distributions and functional distinctions of these components. The N1 component was measured at C3 (re-referenced to Fz), reflecting early somatosensory processing contralateral to stimulation, while the N2 and P2 components, known for their robust vertex-centered distribution and association with affective and cognitive aspects of pain, were measured at Cz (re-referenced to the average of bilateral mastoids). Statistical comparisons were then conducted on the amplitudes and latencies extracted from these specific electrodes.

For Experiment 4 (now designated as Supplementary Experiment), which focused on spontaneous brain oscillations, a different approach was employed. Given that the precise brain regions for the modulatory effects of sensorimotor synchronization on spontaneous EEG are not yet well-established, a data-driven, whole-brain analysis was utilized to explore potential effects across the scalp. As detailed in the Methods section of Supplementary Experiment, we performed one-way repeated-measures ANOVA at each electrode to examine changes in PSD for each frequency band and the changes in aperiodic exponent. To control the Family-Wise Error Rate across all multiple comparisons, the Holm-Bonferroni procedure was applied.

Subsequently, significant clusters of spatially adjacent electrodes showing consistent effects were identified and reported.

We have revised the corresponding sentence in the revised manuscript to more accurately reflect this procedure for Experiments 3 (see lines 283-292), and the methods for original Experiment 4 are now clearly described within the Supplementary Experiment (see Supplementary Information, lines 77-83).

9. *“Although the drumming groups also outperformed the no drumming groups in reducing unpleasantness, the main effect of drumming did not reach statistical significance ($F(1,76) = 3.67, P = 0.059, \eta^2 = 0.05$).” – This statement is incorrect. There are no statistical trends or effects approaching significance. If the results is not significant then the drumming groups did NOT outperform the no-drumming groups.*

Reply: We thank the reviewer for pointing this out. We agree that the original statement was misleading, as the reported effect did not reach statistical significance. We have now removed this inappropriate phrasing from the manuscript and replaced it with a more accurate description of the result (see lines 347-348).

10. *“the more participants felt synchronized and relied on rhythmic auditory sequence, the greater their reductions in pain ratings” – Importantly, the felt synchrony and the actual synchrony (objectively measured) is not necessarily aligned. E.g., Matthews et al., 2022, Music Perception: “Perceived synchrony showed a stronger relation with groove ratings than measured synchrony”. This is just another example of why objectively measured accuracy and stability of drumming need to be reported.*

Reply: As acknowledged in the response to the first main issue, we agree on the importance of objective sensorimotor timing data and recognize its absence as a limitation (see lines 566-573). However, in this study, we focused on the role of subjective experience of synchrony (i.e., perceived synchrony), rather than objective temporal alignment. This emphasis is well-justified

by a growing body of literature, including the Matthews et al. (2022) study cited by the reviewer, which demonstrates that perceived synchrony often holds a stronger predictive power for affective and motivational outcomes (like "groove" or pleasure) than objectively measured synchrony. Given that pain is a complex, multi-dimensional experience influenced heavily by affective and cognitive factors, we specifically aimed to investigate the subjective alignment's role in pain modulation. Indeed, our findings revealed a robust association: participants' felt synchrony was significantly associated with reductions in both pain intensity ($\rho = -0.53$, $P < 0.001$) and unpleasantness ($\rho = -0.27$, $P = 0.028$). This suggests that the subjective experience of temporal alignment is a critical factor in the observed analgesic effects.

Furthermore, to address the reviewer's concern regarding actual task performance and its potential influence on perceived synchrony, we conducted a comprehensive manipulation check in Experiment 2 (see lines 371-389 and Figure S1). This analysis revealed several key points:

- (1) Participants in the in-phase synchrony condition reported significantly stronger perceived audio-visual synchrony compared to other groups ($H_{(2)} = 34.95$, $P < 0.001$), confirming that our audio-visual manipulation successfully induced distinct subjective experiences.
- (2) Participants across all groups reported comparable levels of satisfaction with their task performance ($H_{(2)} = 4.26$, $P = 0.12$), suggesting that the observed pain relief was not simply a byproduct of differential task ease or general satisfaction.
- (3) Our correlation analyses consistently showed that pain reduction was associated with the degree of perceived synchrony and reliance on auditory cues, rather than overall task performance satisfaction (see Figure 4B).

While we agree that these approaches do not substitute for precise motor-timing analyses, they provide converging support for the behavioral relevance of synchrony.

Reference:

Matthews, T. E., Witek, M. A., Thibodeau, J. L., Vuust, P. & Penhune, V. B. Perceived motor synchrony with the beat is more strongly related to groove than measured synchrony. *Music Percept.* **39**, 423-442 (2022).

11. From Figure 5B, I can roughly extract the mean N2 peaks, which are -12 for in-phase, -6 for anti-phase, and -7.5 for asynchrony. Why is the only reported pairwise comparison the significant difference between in-phase and asynchrony? Given that the peak of anti-phase is even lower (and the variance comparable) there should also be a significant in-phase versus anti-phase difference. This inconsistency has to be reported and explained. Leaving this comparison out is untransparent and cherry-picking.

Reply: We appreciate the reviewer's careful observation and for raising the critical point about transparency in reporting. We acknowledge the initial concern regarding the interpretation of N2 amplitude changes. As suggested by other reviewers (also see our reply to Comment #5) we have re-analyzed the EEG data for Experiment 3 using an updated preprocessing and analysis pipeline.

We would like to clarify that Experiment 3 employed a between-subjects design. To account for individual variability in baseline N2 amplitudes, all group comparisons were conducted based on the pre-post changes (i.e., the difference in N2 amplitude before and after the intervention), rather than raw post-intervention peak values.

Regarding the visualization in Figure 5B, it illustrates the group-level LEP waveforms for each group. The associated topographies depict the spatial distribution of these group-level N2 and P2 components, which were computed by spline interpolation at their respective peak latencies. For the statistical analyses and their representation in the accompanying bar graphs, however, we specifically extracted local peak amplitudes (averaged within ± 20 ms around peak latency) for each participant. While these two methods of amplitude assessment (visually identified peaks from averaged waveforms vs. local peak amplitudes within a window) are highly related, it is possible for slight discrepancies to arise between them, which could contribute to perceived visual differences.

Even with the updated pipeline and based on these change scores, our post hoc comparisons consistently revealed a significant reduction in N2 amplitude only between the in-phase synchrony and asynchrony groups (see Figure R7). Although the anti-phase group's N2 amplitude change might appear visually similar to asynchrony, when considering the inherent

variability within this between-subjects design, the difference between the in-phase and anti-phase groups did not reach statistical significance ($P = 0.058$).

To address the reviewer's concern about transparency and to illustrate this variability, we have now provided individual data points in the revised Figure 5, which clearly depicts the distribution of N2 change scores for each group. This visual representation, alongside the updated statistical analyses, demonstrates that all relevant comparisons, including those that did not reach significance, are fully reported in the revised manuscript (see Figure 5B and lines 430-435).

Figure R7. (A) Average waveforms at Cz electrode (with bilateral mastoid as the reference electrode) of each group in response to nociceptive laser stimuli in the pre-task (solid line) and the post-task (dash line) sessions (black lines were difference waves between two sessions), as well as the scalp topographic maps of N2 (upper) and P2 (lower) components at their respective peak latencies. (B) Comparisons of changes in subjective pain ratings as well as N2 and P2 amplitudes among the in-phase synchrony (green), anti-phase synchrony (blue), and asynchrony (red) groups. Error bar represents \pm one standard error of the

mean. Pre.: the pre-task session; Post.: the post-task session; * $P < 0.05$.

12. What kind of filter was applied to Figure 6A top left? Shouldn't this be unfiltered EEG data?

Reply: The original Figure 6A (top left) was not filtered EEG data but rather a zoomed-in view of specific frequency bands (α - and γ -bands) extracted from the power spectrum, intended to highlight the differences between conditions in those frequency ranges.

To address the reviewer's concern, we have replaced the original Figure 6A with a new Figure S4, presenting the full, unmodified PSD plot across the entire frequency range (see Supplementary Information, Figure S4).

13. Discussion: "pain relief was significantly enhanced when participants' drumming strokes were perfectly synchronized with the auditory drumbeats" – again: this statement does not follow from the data as sensorimotor synchronization was not measured. It is incorrect.

Reply: We agree that, since we did not collect objective measures of sensorimotor synchronization (e.g., tap-to-beat asynchrony), it is inappropriate to state that participants' drumming strokes were "perfectly synchronized" with the auditory cues. In fact, this statement was based on participants' subjective reports of perceived synchrony. Accordingly, we have revised the sentence in the Discussion section to clarify this point and avoid overstatement (see lines 473-475).

14. The whole discussion of Experiment 4 (Chapter 4.3) does not follow from the data and results. It could also be a pure effect of movement versus rest. With the current design it is impossible to make any claim about the specific effect of sensorimotor synchronization.

Reply: We agree that, with the current design, Experiment 4 (now designated as a Supplementary Experiment) cannot distinguish the effects of sensorimotor synchronization from those of movement *per se*, as it only included in-phase drumming conditions and a resting

baseline.

In response to this critical point, and in light of its move to Supplementary Information, we have thoroughly revised the discussion. The main manuscript now only briefly references its exploratory findings in the main manuscript and directs readers to Supplementary Information for a comprehensive discussion (see lines 552-561).

Within Supplementary Experiment, we explicitly acknowledge this design limitation. We state that while the observed changes in spontaneous γ -band activity may reflect neural modulations associated with active rhythmic engagement, we do not present them as conclusive evidence for a synchronization-specific effect within the main narrative.

We further discuss how certain characteristics of the observed γ -band enhancement, which differ from typical transient, localized movement-induced γ -band activity (as described by Cheyne et al., 2008; Crone et al., 1998), might suggest potential involvement of broader cognitive processes beyond mere motor output. However, we emphasize that without active non-synchronous movement control, such interpretations remain speculative and require further investigation. This approach avoids overinterpretation and fully respects the design's constraints (see Supplementary Information, lines 98-100, and 145-149).

References:

Cheyne, D., Bells, S., Ferrari, P., Gaetz, W., & Bostan, A. C. Self-paced movements induce high-frequency gamma oscillations in primary motor cortex. *Neuroimage* **42**, 332-342 (2008).

Crone, N. E., Miglioretti, D. L., Gordon, B., & Lesser, R. P. Functional mapping of human sensorimotor cortex with electrocorticographic spectral analysis. II. Event-related synchronization in the gamma band. *Brain* **121**, 2301-2315 (1998).

Minor issues:

15. Page numbers are missing.

Reply: As suggested, we have now added page numbers throughout the revised manuscript.

16. Only the number of female participants is reported. This does not mean that the other participants were male. What about non-binary or other participants, or participants who would prefer not to provide this detail? Were these options provided?

Reply: As suggested, we have now added the number of male participants for each experiment in the revised manuscript to provide complete gender information (see lines 113, 204, and 243; Supplementary Information, line 30).

17. The design of Experiment 1 should mention that it was a between-subject design with four groups.

Reply: As suggested, we have revised the description of Experiment 1 to explicitly state that it employed a between-subjects design with four groups (see line 164).

18. The following sentences are unclear to me: “Given the pre-post experimental design, any group differences in the post-test, relative to the pre-test, cannot be attributed to attentional distraction during the intervention session. Instead, they reflect a more direct and uncontaminated measure of the aftereffects of different experimental manipulations.”

Reply: We understand that the original phrasing may be unclear. In each group, we conducted pain rating tasks both before and after the intervention, using a series of medium-intensity laser stimuli followed by participant ratings. This pre-post design allowed us to assess changes in pain perception that are more likely attributable to the effects of the intervention itself, rather than other transient factors during the task. To clarify, we have revised the relevant sentence in the manuscript (see lines 182-186).

19. Experiment 2, participants: “ANOVA focusing on fixed effects across three groups” – Here it should be mentioned whether this is a within or between design.

Reply: We apologize for the confusion. Consistent with Experiment 1, both Experiments 2 and 3 employed a between-subjects design, in which participants were randomly assigned to one of three groups. Therefore, ANOVA focused on fixed effects across independent groups. We have clarified this point in the revised manuscript to avoid misunderstanding (see lines 226-229, and 249-250).

20. Experiment 3: “Group” is a confusing term for the within-subjects design. Maybe “condition”?

Reply: As noted in our response to the previous comment, Experiment 3 employed a between-subjects design rather than a within-subjects design. Therefore, the use of the term “group” is appropriate in this context. Nevertheless, we have carefully reviewed and revised the relevant text to ensure that the study design is clearly stated (see lines 249-250).

21. “drumming to music, a representative form of groove” – drumming to music can, under specific circumstances, induce an experience of groove, but drumming to music per se is not a representation of groove.

Reply: We agree that our original phrasing was imprecise. In accordance with the reviewer’s suggestion, we have revised the sentence in the manuscript (see lines 512-514).

Reviewer #3 (Remarks to the Author):

This is a very impressive series of four experiments, with a total sample size of 244 participants. Understanding mechanisms underlying music induced analgesia is an important and timely topic. Not a lot of studies on sensorimotor synchronization but there are a few studies. The authors already cite Werner et al. (2023), but there is also another study that just came out that I think could be interesting to them (https://journals.lww.com/pain/fulltext/9900/individualizing_musical_tempo_to_spontaneous_rates.810.aspx).

In any case, this is a topic of growing relevance, and the current set of studies represents by far the most thorough and well-designed exploration to date. The findings convincingly demonstrate that sensorimotor synchronization yields stronger analgesic effects than either motor or auditory stimulation alone, and that in-phase synchronization is more effective than anti-phase or asynchronous conditions. The evidence pointing to the mediation of these effects by the N2 component of the ERP is particularly compelling—very nicely done.

That said, I do have some concerns about Study 4. This could be put in limitation, but study 4 could also be removed if other reviewers or editor feel the same way about it:

Reply: We sincerely thank the reviewer for the generous and encouraging comments on our work. We are also grateful for the recommendation of the recent study by Yi et al. (2025), which provides valuable insights into individualized musical tempo and its relevance to pain perception. We have reviewed the article and have now cited it in the revised manuscript (see lines 51-57).

Regarding the concerns about Experiment 4, and after carefully considering the feedback from other reviewers regarding its design limitations and the interpretability of its findings within the main narrative, we have decided to move the entire content of Experiment 4 to Supplementary Information. This allows us to retain its valuable, albeit exploratory, neurophysiological insights into spontaneous brain oscillations, while ensuring the main manuscript remains sharply focused on the core behavioral and laser-evoked potential findings from Experiments 1-3. We have also explicitly discussed its methodological limitations within

the Supplementary Information and adjusted related interpretations in the main discussion to be more cautious and speculative.

A point-by-point response to the comments follows.

Reference:

Yi, W., Palmer, C., Serian, A., & Roy, M. Individualizing musical tempo to spontaneous rates maximizes music-induced hypoalgesia. *Pain* **10**, 1097 (2022).

1. The “rest” condition always occurs in the middle of the sequence and is not counterbalanced. This undermines any comparison involving the rest/silence condition and casts doubt on the conclusions drawn from Study 4.

Reply: We acknowledge that in Experiment 4, the rest condition was always presented in the middle of the sequence and was not counterbalanced. However, this design choice was intentional and guided by the specific goals of our within-subject paradigm. The rest condition served as a controlled baseline period between the two active sensorimotor synchronization blocks, minimizing potential carryover effects and allowing us to isolate the neural changes following sensorimotor activity. Importantly, our main focus was not to directly compare the rest condition with the active synchronization conditions, but rather to examine the within-subject neural dynamics before and after active engagement.

Thus, while we fully acknowledge that the fixed position of the rest condition represents a design limitation for certain types of comparisons, we believe it fulfilled a consistent and functionally necessary role in facilitating our primary exploratory aims.

Both the rationale behind this design choice and its potential implications for the interpretation of our findings are thoroughly discussed within Supplementary Experiment (see Supplementary Information, lines 48-50, and 156-159).

2. The sample size ($n = 20$) is quite small relative to the other experiments. Could the authors

clarify how this sample size was determined?

Reply: The sample size for Experiment 4 ($n = 20$) was smaller than in the other experiments because it employed a within-subject design, in which each participant completed all three conditions, while others employed a between-subject design. This design increases statistical power by reducing between-subject variability. The sample size was determined *a priori* using G*Power software, with the statistical power set at 0.8, alpha at 0.05, and an expected effect size ($f = 0.4$) based on prior work by Werner et al. (2021). The analysis indicated a minimum required sample size of 12 to achieve adequate statistical power. To account for potential data loss and variability, we recruited 20 participants. This information has been made clear in Supplementary Experiment (see Supplementary Information, lines 33-35).

Reference:

Werner, L. M., Skouras, S., Bechtold, L., Pallesen, S. & Koelsch, S. Sensorimotor synchronization to music reduces pain. *PLoS One* **18**, e0289302 (2023)

3. Why couldn't the frequential EEG analyses be performed on the participants from Study 3? This larger sample might have offered more statistical power.

Reply: We thank the reviewer for this insightful suggestion. Experiment 3 was designed to investigate the effect of sensorimotor synchronization on pain perception using nociceptive-evoked EEG responses (i.e., LEPs), rather than spontaneous brain activity. In Experiment 3, no enough EEG data at rest were collected that would allow us to isolate pre- and post-intervention brain oscillations for spectral EEG analyses, as those implemented in Experiment 4.

A few minor suggestions:

4. It would be helpful to briefly justify the inclusion of EEG measures in the introduction.

Reply: As suggested, we have added a brief paragraph in the Introduction section to justify the inclusion of EEG measures in our study. Specifically, we highlight that EEG offers a

noninvasive and temporally precise means of capturing neural responses, which is crucial for investigating the dynamic cortical mechanisms underlying pain modulation. We elaborate that EEG allows us to capture pain-evoked neural responses, such as laser-evoked potentials (LEPs). These LEPs are considered core neural correlates mediating music- and rhythm-based analgesia. We have revised the relevant paragraph in the Introduction accordingly (see lines 81-86).

5. I'm curious whether there were any differences between the 90 bpm and 120 bpm conditions. The recent study by Yi et al. (2025) suggests that 120 bpm may be more effective for most participants.

Reply: Indeed, the study by Yi et al. (2025) identified an optimal tempo range corresponding to inter-tap intervals (ITIs) between 256 and 722 ms (approximately 83–234 bpm), within which both our selected tempi (i.e., 90 bpm and 120 bpm) fall. Although Yi et al. did not report the group mean or median for spontaneous motor tempo, their Figure 2 (see below) suggests that most participants had average spontaneous production rates (SPR) between 350 and 550 ms, corresponding roughly to 110–170 bpm. This implies that 120 bpm may be closer to the individual optimal tempo for many participants in that sample.

However, in our study, we did not observe significant differences in pain modulation or neural responses between the 90-bpm and 120-bpm conditions. One possible reason is that both tempi fall within a comfortable and easy to synchronize, making them equally effective in promoting sensorimotor coupling and associated analgesic effects. Another consideration is that Yi et al. (2025) suggested that the internal rhythmic oscillator reflected by each participant's SPR may play a key role in determining the optimal tempo for pain modulation. The alignment between external rhythm and internal oscillatory timing could be more important than the absolute tempo itself in inducing analgesia. Therefore, it is possible that both 90 bpm and 120 bpm may have sufficiently matched many participants' intrinsic rhythmic preferences to elicit comparable effects.

This point has been clarified in Supplementary Experiment (see Supplementary Information, lines 117-127).

[figure redacted]

Figure 2 from Yi et al. (2022) published in PAIN

Reference:

Yi, W., Palmer, C., Serian, A., & Roy, M. Individualizing musical tempo to spontaneous rates maximizes music-induced hypoalgesia. *PAIN* **10**, 1097 (2022).

6. It would be great to visualize the mediation pathway more clearly in the figure by including the $a*b$ path and indicating whether the mediation effect is statistically significant.

Reply: As suggested, we have revised the mediation diagram to explicitly display the a , b , and $a*b$ paths, along with corresponding path coefficients (see Figure R8). We have now annotated the figure to indicate the statistical significance of the mediation effect (see Figure 5).

Figure R8. The mediation model. Changes in N2 amplitude mediated the effect of group on pain

intensity ratings. The coefficients for each path are presented. ** $P < 0.01$; * $P < 0.05$.

7. One conceptual point worth at least acknowledging: is it possible that the superior effects of the sound + drumming and in-phase conditions reflect the fact that these are simply easier tasks? Synchronizing to an auditory cue may be more intuitive than to a visual one. If so, the observed effects might reflect task ease rather than sensorimotor coupling per se. Performance data (e.g., objective synchrony measures) could help clarify this. If there is a difference in performance, then perhaps that means that the effect is driven by task ease, not sensorimotor synchronization per se. For instance, we could predict to obtain similar results with chess puzzles of differing difficulties. While this is potentially important for the field in the long term, I think that this is more a question for another study. But if the authors agree, this is something that they could potentially discuss in the discussion section of the paper.

Reply: We appreciate the reviewer's comment on the potential influence of task difficulty. It is indeed possible that the task in the in-phase group was more natural or less cognitively demanding than those in the other two groups. However, we believe that the observed effects are less likely to be solely driven by general task ease, as our manipulation check, now detailed with additional behavioral analyses in Experiment 2 (see lines 371-389 and Figure S1), provides important insights.

Specifically, participants across all groups similarly attempted to follow the visual flashing-dot cues as instructed ($H_{(2)} = 3.11$, $P = 0.21$), indicating consistent effort. Crucially, while the in-phase group reported significantly stronger perceived audio-visual synchrony ($H_{(2)} = 34.95$, $P < 0.001$), confirming that our manipulation successfully induced different levels of perceived synchrony as intended, all participants reported comparable levels of task performance satisfaction ($H_{(2)} = 4.26$, $P = 0.12$). This suggests that the observed pain relief was not simply due to differential task ease or general satisfaction across groups, as participants perceived similar subjective success despite varying synchrony conditions.

Furthermore, our correlation analyses shown in Figure 4B further supported that pain reduction was associated with the degree of perceived synchrony (pain intensity: $\rho = -0.53$, $P < 0.001$;

unpleasantness: $\rho = -0.27, P = 0.028$) and reliance on auditory cues (pain intensity: $\rho = -0.39, P = 0.001$; unpleasantness: $\rho = -0.25, P = 0.040$), rather than general task performance satisfaction (see Figure R9). This suggests a possible mechanism related to perceived sensorimotor synchrony, beyond task ease (see lines 525-528).

Nonetheless, we acknowledge that the in-phase condition, involving congruent and intuitive sensorimotor coupling, might have been perceived as more natural. This naturalness could indeed contribute to greater engagement and potentially influence pain perception through pathways like mood enhancement. We have now discussed this issue in the revised manuscript (see lines 566-573).

Figure R9. Correlations between changes in pain ratings (both pain intensity and unpleasantness) and self-reported evaluations (Q1: the extent to which the drumbeats and flashing dots were synchronized with each other; Q2: the extent to which you were playing the drum in response to drumbeats; Q3: the extent to which you were playing the drum in response to flashing dots, and Q4: the extent to which you were satisfied with your task performance) *** $P < 0.001$; ** $P < 0.01$; * $P < 0.05$.